# Varying Manifolds in Diffusion:
## From Time-varying Geometries to Visual Saliency

## Abstract

Building on the manifold hypothesis, which suggests that generative models learn data distributions residing on low-dimensional manifolds, this paper investigates the time-varying manifold sequence induced by the generation process through the lens of differential equations in diffusion models. Our primary contribution is the introduction of the *generation rate*, a novel metric that quantifies local manifold scaling over time. For image data, we show that the accumulated generation rate, referred to as the *generation curve*, strongly correlates with intuitive visual properties, such as the saliency of image components. By leveraging modifications to the generation curves, we propose a unified framework for a range of image manipulation tasks, including semantic transfer, object removal, saliency adjustment, and image blending. Comprehensive evaluations, supported by both the qualitative and quantitative results, highlight the effectiveness of our framework across these diverse tasks.

## 1. Introduction

The variation of natural data, i.e., how the data can change, is governed by several continuous variables, such as lighting and colors in images. Consequently, the data exhibit fewer degrees of freedom (dimensions) than their ambient space, exemplified by the total pixels of natural images. This observation inspires manifold hypothesis (Tenenbaum et al., 2000; Roweis & Saul, 2000), which states that real-world data are conceptualized as points on or near a low-dimensional manifold within a high-dimensional ambient space.

Generative models, such as Variational Autoencoders (VAE) (Kingma & Welling, 2014) and Generative Adversarial Networks (GANs) (Goodfellow et al., 2014), seek to learn the statistical data distribution by constructing the mapping from a simple base distribution, such as a Gaussian distribution, to the data distribution. The learned distribution inherently encodes the underlying data manifold. Recently, diffusion models or score-based models (Sohl-Dickstein et al., 2015; Ho et al., 2020) have substantially enhanced

the expressiveness of generative modeling by representing the distribution transformation process as a stochastic differential equation (SDE). Inspired by natural diffusion phenomena, diffusion models elucidate how data distributions evolve under noise perturbations.

Data manifolds provide a geometric perspective for data analysis. On one hand, viewing data as points on a curved manifold enables effective examination of pairwise geometric distances and localized variations among samples. (Arvanitidis et al., 2018; Miolane et al., 2019). On the other hand, the geometry of these manifolds can reflect domain-specific attributes, such as the connectivity of graph-structured data (Topping et al., 2022) or complexity and classification characteristics of images (Tempczyk et al., 2021; Baptista et al., 2024). In the context of diffusion models, the generative process based on differential equation offers insights into the evolution of data manifolds. The analysis of time-varying data manifolds enables a more comprehensive understanding of how data is generated and how its underlying geometric structure changes over time.

In this paper, we propose a metric to gauge the rate of change of the data manifold in diffusion models as a function of time. As our key observation, we show that such a change rate corresponds to the rate of information removal during the diffusion process, or the rate of information injection during the reverse generation process. Therefore, in the reverse process, we call our metric the *"generation rate"*, which changes over time to define the *"generation curve"*. Furthermore, we demonstrate that the generation curve effectively captures the visual properties of image data. Utilizing this connection, we construct a unified framework for a range of image manipulation tasks, including *semantic transfer, object removal, saliency manipulation*, and *image blending*, by manipulating the shape of the generation curve using stochastic optimization. Finally, through comprehensive evaluation, we show that our framework consistently outperforms the existing approaches in performing these image manipulation tasks.

## 2. Background

It is widely accepted that the distributions of high-dimensional observed data reside on a lower-dimensional

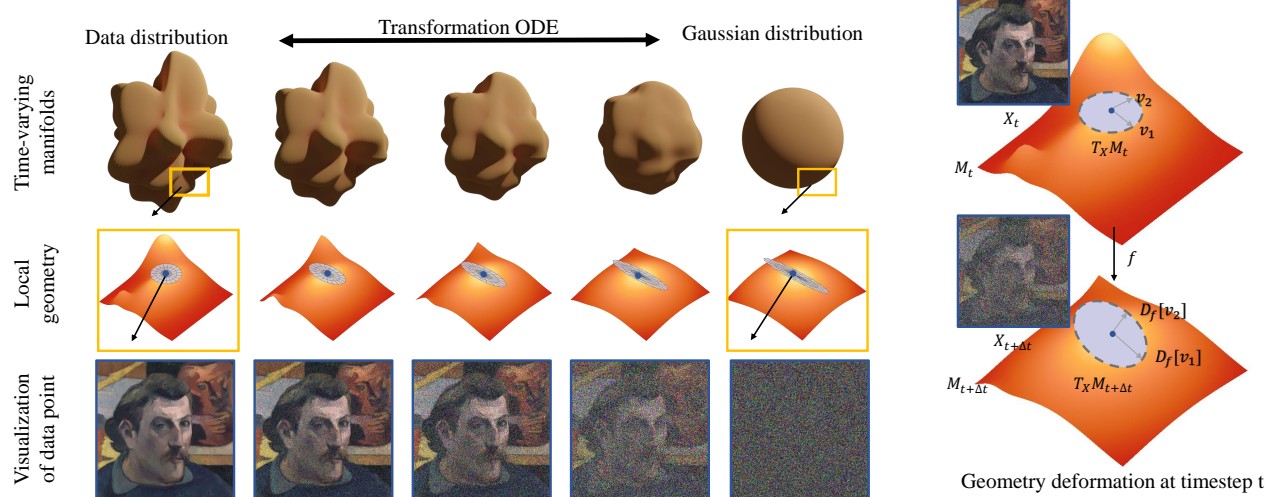

Figure 1. We consider the data manifold at each time level of the transformation Equation 3 (top row), and analyze the local geometry (second row) around a nominal sample (third row). The notations used to define the local geometry are illustrated on the right.

manifold $M$ embedded in ambient space $R^d$ (Tenenbaum et al., 2000; Roweis & Saul, 2000). The notion of data manifold not only offers a low-dimensional representation of the data but also proves highly useful for data analysis and processing (Tempczyk et al., 2021; Topping et al., 2022; Baptista et al., 2024). In this paper, we adopt basic concepts from differential geometry, including the tangent space $T_x M$ as a linear approximation of the manifold at a point $x$ and differential mappings that act on this tangent space. More background details are presented in Appendix A.

### 2.1. Diffusion Process as SDE

The diffusion model (Sohl-Dickstein et al., 2015; Ho et al., 2020) is a type of stochastic generative model that gradually adds noise to the original data in a forward diffusion process and generates realistic data samples via a reverse denoising process. It can be formulated as stochastic differential equations (SDEs) (Song et al., 2021b) with a continuous time variable $t \in [0, T]$. The forward diffusion process, which evolves a probabilistic distribution towards a more uniform or stable state over time through random perturbations, is:

$$dX_t = \mu(X_t, t)\, dt + \sigma(X_t, t)\, dW_t, \quad (1)$$

where $X_t$ represents the state of the process at time $t$, $\mu(X_t, t)$ is the drift coefficient, $\sigma(X_t, t)$ is the volatility coefficient, and $dW_t$ is the differential of a Wiener process.

The reverse SDE, used for denoising and generating data, is formulated as:

$$dX_t = [\mu(X_t, t) - \sigma^2(X_t, t)\nabla_x \log p_t(X_t)]\, dt + \sigma(X_t, t)\, dW_t, \quad (2)$$

where $\nabla_x \log p_t(X_t)$ is the score function of the probability density function $p_t(X_t)$.

We can further derive a deterministic process with trajectories that share the same marginal probability densities as the SDE (Equation 1). This is formulated as an ordinary differential equation (ODE) (Song et al., 2021b):

$$dX_t = [\mu(X_t, t) - \frac{1}{2}\sigma^2(X_t, t)\nabla_x \log p_t(X_t)]\, dt, \quad (3)$$

In this paper, we adopt this deterministic approach and use its specific discrete form from (Song et al., 2021a):

$$\frac{X_{t-\Delta t}}{\sqrt{\alpha_{t-\Delta t}}} = \frac{X_t}{\sqrt{\alpha_t}} + \left(\sqrt{\frac{1-\alpha_{t-\Delta t}}{\alpha_{t-\Delta t}}} - \sqrt{\frac{1-\alpha_t}{\alpha_t}}\right)\epsilon_\theta^t(X_t), \quad (4)$$

where $\alpha_t$ is a time-dependent variable as defined in (Song et al., 2021a) and $\epsilon_\theta^t(X_t)$ is a neural network with parameter $\theta$ trained to approximate noise, which is related to the score function by reparametrization $\nabla_x \log p_t(X_t) = -\epsilon_\theta^t(X_t)/\sqrt{1-\alpha_t}$ (Song et al., 2021b). In this form, one can directly and deterministically obtain a predicted $\hat{X}_0$ from the $X_t$ by: $\hat{X}_0(X_t) = (X_t - \sqrt{1-\alpha_t}\epsilon_\theta^t(X_t))/\sqrt{\alpha_t}$, which can further be considered as an estimation of the generation state.

### 2.2. Time-Varying Manifolds in Diffusion Models

Within the realm of generative models, the learned data distribution offers a natural foundation for studying data geometry. In particular, the ODE-based formulation of diffusion models suggests that their generative process does more than simply denoising individual data points: it induces a family of diffeomorphisms on the data manifold. This continuous transformation can be discretized into a sequence of manifolds $\{M_t\}$, as illustrated in Figure 1.

In this paper, we adopt a geometric viewpoint on how data evolve through the generation process, linking data

attributes to manifold geometry. A challenge lies in accessing geometric quantities such as the local tangent basis. Some existing work (Batzolis et al., 2022) derives the manifold's tangent space by exploiting gradient information from score-based (diffusion) models, but this approach is computationally prohibitive for high-dimensional data. Alternatively, building on empirical evidence that intermediate neural network spaces tend to compress normal directions while preserving the tangent directions (Kvinge et al., 2024), (Park et al., 2023) propose to approximate the tangent basis of a data manifold within a diffusion model's U-Net architecture. Specifically, they regard the encoder layers of the noise predictor (U-Net) as a compression map $h_t$. In practice, they apply the power method to the differential $D_{h_t}$ of $h_t$, which extracts the leading right singular vectors to approximate the manifold's local tangent basis. We adopt this approach to derive the tangent space of $M_t$.

## 3. Generation Rate & Generation Curve

In this section, we first define our notation of generation rates and generation curves in diffusion models (Section 3.1). We then demonstrate, using a pre-trained image diffusion model, that these curves effectively capture how quickly the model generates visual content. In particular, we show that the fluctuations of these curves exhibit a strong correlation with visual saliency (Section 3.2).

### 3.1. Definition

The key idea of our analysis lies in the temporally local analysis of the time-varying manifolds $\{M_t\}$. Between consecutive manifolds, we define the forward diffusion mapping as $X_t = f_t(X_{t-\Delta t})$, and the reverse diffusion mapping as $X_{t-\Delta t} = f_t^{-1}(X_t)$. These manifold-to-manifold transformations induce differential mappings $D_{f_t}$ and $D_{f_t^{-1}}$ on the corresponding tangent spaces $T_x M_t$, transforming a tangent vector from one manifold to the next. Following prior work (Bengio et al., 2013; Hanin & Rolnick, 2019), we use local scaling of the tangent space to track how a manifold deforms. In particular, the scaling of a tangent vector refers to the change in its length before and after the differential mapping, thereby quantifying the local geometric deformation at each timestep, as illustrated in Figure 1.

An empirical finding is that, under the parametrization in Equation 4, the singular values of $D_{f_t}$ for tangent vectors almost always fall in the range $(0, 1)$, implying that $f_t$ is a contracting mapping when restricted to $M_t$. This is consistent with the fact that the diffusion process removes information from the data and injects entropy into the distribution on $M_t$ as $t$ increases. Similarly, we can consider the reverse process and the associated map $f_t^{-1}$, whose differential $D_{f_t^{-1}}$ empirically has singular values in the range $(1, \infty)$. This corresponds to injecting information into the distribution

by reducing its entropy. Based on these observations, we define our information *generation rate* as the norm of the directional derivative:

$$\|D_{f_t^{-1}}(X_t)[v]\| : T_x M_t \mapsto R, \tag{5}$$

along a unit tangent-space variation $v$.

A potential pitfall of Equation 5 lies in the requirement that $v$ is a tangent-space variation. In practice, however, an arbitrarily sampled variation $v \in R^d$ to a noised image $X_t$ might not lie in the tangent space $T_x M_t$. To mitigate this flaw, we define the projection operator $\text{Proj}(v)$ as the projection of $v$ onto the tangent space spanning by the leading singular vectors, whose derivation is explained in Section 2.2. Specifically, we define the *projected generation rate* as:

$$r_t(X_t, v) = \|D_{f_t^{-1}}(X_t)[\text{Proj}(v)]\|, \tag{6}$$

which enables us to use an arbitrary variation $v$ in the ambient space $R^d$. The variation vector $v$ provides an important additional degree of freedom: for example, by choosing $v$ to represent a specific image component, e.g. a single pixel or a channel of a pixel, we can measure the generation rate specifically for this component.

Using the diffusion ODE (Equation 4), we can uniquely trace a reversible path $\mathcal{X} = \langle X_0, X_1, ..., X_T \rangle$ either forward (from $X_0$) or backward (from $X_T$). This sequence is also denoted as $\mathcal{X}(X_0)$ or $\mathcal{X}(X_T)$. We define the *generation curve* $c(\mathcal{X}, v)$ as the discrete sequence $\langle r_t(X_t \in \mathcal{X}, v) \rangle_{t=0}^T$, calculated at each time step from $t = T$ down to $t = 0$ for a fixed variation $v \in R^d$ and a given path $\mathcal{X}$.

### 3.2. Connection to Visual Saliency

In this section, we present our comprehensive analysis to demonstrate that the generation curve is strongly connected to the visual properties of images. For simplicity, we consider single-channel images since additional channels can be simply treated as extra dimensions. We set the vector $v$ as a unit vector $v = e_{ij}$ that takes value 1 at $ij$-th pixel and zero otherwise. Then, we project it to the tangent space of the image manifold to compute its generation rate with Equation 6. This setting allows us to investigate the rate of information generation for a single pixel.

To extend the notion of generation rate to an image patch $\mathcal{A}$ containing multiple pixels $\{e_{ij}\}$, we slightly abuse notation and define it as the following average:

$$r_t(X_t, \mathcal{A}) = E_{e_{ij} \sim U(\mathcal{A})}[r_t(X_t, v(e_{ij}))], \tag{7}$$

where $v(e_{ij})$ denotes the vectorization of pixel $e_{ij}$ and $U(\cdot)$ is the uniform distribution.

In Figure 2, we plot the generation curves for a column or row of image pixels, with each curve corresponding to an

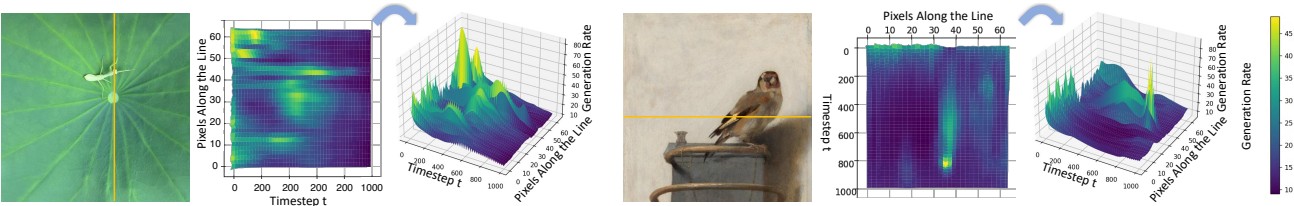

Figure 2. Generation curves for a column or row of image pixels (yellow). The generation curves fluctuate significantly at the pixels with high visual saliency, such as the wing tip of the bird.

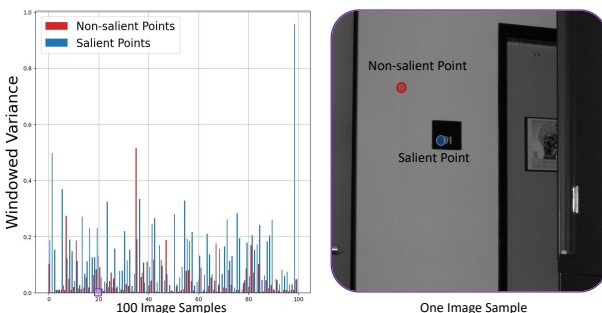

Figure 3. Visual saliency analysis. Left: Curve fluctuation statistics comparing salient (blue) and non-salient (red) pixels across image samples. Right: An example image from the test set.

individual pixel. We observe that pixels exhibiting high visual saliency, such as those in the bird's wing tip and body, demonstrate pronounced fluctuations (peaks) in their generation curves. Conversely, in lower-saliency regions the curves remain notably smoother, except for a sharp rise when $t$ approaches 0. We further conduct a large-scale analysis by using 100 pictures from the visual saliency dataset (Borji & Itti, 2015). For each image, we sample one salient pixel and one non-salient pixel from the ground truth. For both pixels, we take the windowed variance: $c(\mathcal{X}, v(e_{ij})) \rightarrow R$ to measure the fluctuation of its generation curve, and the statistics are shown in Figure 3 (Appendix C.1 for more details). For 86% of the images, higher visual saliency leads to higher fluctuation, validating the high consistency between curve fluctuation and visual saliency.

For other morphological factors of the curve, such as the position and curvature of the peaks, we experimentally found that these are determined by more specific and low-level visual properties of the underlying object. For instance, image patches with different materials like grassland and woodland often exhibit different curve shapes, although they share similar visual saliency.

We also notice an alternative way (Choi et al., 2022; Kwon et al., 2023) to consider the generation rate in diffusion models. For each noised image $X_t$, we can compare the visual similarity of its predicted $\hat{X}_0$ (Song et al., 2021a) to the real image $X_0$, e.g., by computing the perceptual distance (Zhang et al., 2018; Caron et al., 2021) between the two images. Such computation can be performed with respect to an image patch by applying a region mask. We

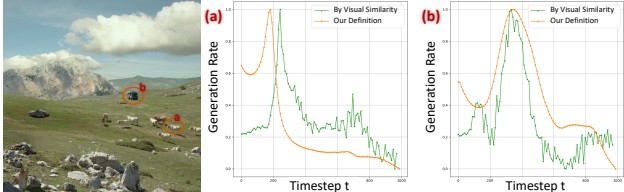

Figure 4. Comparison between perceptual-based curves (green) and our generation curves (orange). The curves correspond to two objects indicated in the specified regions of the left image.

consider this similarity as the generation state, and the time derivative of this state can also be interpreted as a generation rate. In Figure 4, we compare the generation curve computed using our definition and this alternative definition for objects (details in Appendix C.2). We notice that the two curves exhibit similar trends while our curve has much less noise. Besides, due to the nature of visual metrics, this alternative method is applicable only to regions with semantically complete objects and often exhibits heavy noise or even negative generation rates in areas with less prominent visual features, especially with background patches. These limitations prevent this alternative method from providing reliable indications of the information generation rate, limiting its usage in analyzing the generation process.

## 4. Curve Matching

Since the generation curve is intrinsically related to certain visual properties of images, it enables the manipulation of those properties by modifying image curves. In this section, we propose a curve matching algorithm that adjusts the curves via an optimization procedure.

### 4.1. Approximate Computation of Generation Curve

To calculate and modify the curves $c(\mathcal{X}, v)$, we must evaluate the generation rate at each time step. This computation, according to Equation 6, involves the projection operation Proj$(v)$ for image pixel vectors, which depends on a power-method-based derivation of the tangent space (see Section 2.2). Since the power method is not differentiable, the projection Proj$(v)$ impedes gradient-based optimization. Therefore, we seek for a substitute, i.e. a differentiable component of the generation curve that inherits its correlation with visual properties. Using Equation 4, we can rewrite

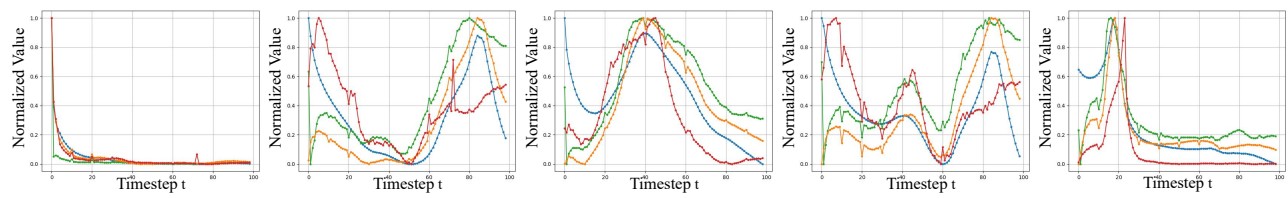

Figure 5. The generation curves approximated by four different methods: $\|D_{f_t^{-1}}(X_t)[\text{Proj}(v)]\|$ •, $\|D_{\epsilon_\theta^t}(X_t)[\text{Proj}(v)]\|$ •, $\|D_{h_t}(X_t)[\text{Proj}(v)]\|$ •, $\|D_{h_t}(X_t)[v]\|$ •, normalized into range $[0, 1]$.

differential of the reverse diffusion mapping $D_{f_t}^{-1}$ as:

$$D_{f_t^{-1}}(X_t)[v] = A(t)v + B(t)D_{\epsilon_\theta(X_t)}(X_t)[v], \quad (8)$$

where $A(t) = \sqrt{\alpha_{t-\Delta t}/\alpha_t}$ and $B(t) = (\sqrt{1 - \alpha_{t-\Delta t}} - \sqrt{\alpha_{t-\Delta t}(1 - \alpha_t)/\alpha_t})$. Substituting $\text{Proj}(v)$ for $v$, the square of generation rate $r_t(X_t, v)$ takes the form:

$$\begin{aligned} r_t^2(X_t, v) &= \langle D_{f_t^{-1}}(X_t)[\text{Proj}(v)], D_{f_t^{-1}}(X_t)[\text{Proj}(v)] \rangle \\ &= A^2(t)\|\text{Proj}(v)\|^2 + B^2(t)\|D_{\epsilon_\theta^t(X_t)}(X_t)[\text{Proj}(v)]\|^2 \\ &\quad + A(t)B(t)\langle \text{Proj}(v), D_{\epsilon_\theta^t(X_t)}(X_t)[\text{Proj}(v)] \rangle, \quad (9) \end{aligned}$$

where $\langle \cdot \rangle$ denotes canonical Euclidean inner product. Empirically, as demonstrated in Figure 5, the curve of $\|D_{\epsilon_\theta^t}(X_t)[\text{Proj}(v)]\|$ dominates the overall shape of the curve of $r_t$. Hence, in practice, we only retain this component in our generation rate formulation, allowing us to control the resulting generation curves and thereby adjust the corresponding visual properties of the images.

To circumvent the non-differentiable projection, we leverage the bottleneck (encoder) mapping $h_t$ (Section 2.2) of the network $\epsilon_\theta^t$. As observed in (Shwartz-Ziv & Tishby, 2017; Saxe et al., 2018), bottleneck layers often exhibit similar singular value distributions to those of the final outputs, implying a shared directional scaling (i.e. how different directions are stretched or compressed). This similarity in scaling implies the similarity in accumulated curves, enabling the curve of $\|D_{h_t}(X_t)[\text{Proj}(v)]\|$ to serve as a substitute for the curve of $\|D_{\epsilon_\theta^t(X_t)}(X_t)[\text{Proj}(v)]\|$. Moreover, since $D_{h_t}$ already compresses normal directions to derive the tangent space, it eliminates the need for $\text{Proj}(\cdot)$. Consequently, the curve of $\|D_{h_t}(X_t)[v]\|$ becomes a natural alternative to $\|D_{\epsilon_\theta^t(X_t)}(X_t)[\text{Proj}(v)]\|$. Figure 5 compares several curves obtained by these alternatives. Despite numerical differences across the curves, their similar shapes meet our needs for manipulating images. Since $\|D_{h_t}(X_t)[v]\|$ is readily optimizable via gradient descent, we adopt it in the subsequent image manipulation applications.

### 4.2. Optimization Algorithm

The proposed curve matching algorithm manipulates the visual properties of an image patch by aligning the shapes of curves. Its input contains an image $X_0$ with its associated noisy sequence $\mathcal{X}$, a specified reference generation curve

$c^\star$ derived from pixel $p^\star$ with the corresponding reference generation rates $r_t^\star$, and a source image patch $\mathcal{A} \in X_0$ to be edited. Our objective is to uniformly transform the patch $\mathcal{A}$ into the visual content represented by $p^\star$, by modifying the generation curves of its pixels to match $c^\star$.

Specifically, we search for a desired image $\bar{X}_0$ via the following optimization:

$$\text{argmin}_{\bar{X}_0} E_{e_{ij} \sim U(\mathcal{A})}[D(c(\mathcal{X}(\bar{X}_0), e_{ij}), c^\star)], \quad (10)$$

where $D$ is some distance metric. Because each curve value is computed independently at discrete time instances, we use an $L_1$-type distance. Moreover, to focus optimization on the more salient parts of the curve $c(\mathcal{X}, e_{ij})$, thereby better reflecting the crucial distinctions between it and $c^\star$, we adopt an importance sampling strategy. Specifically, we first normalize the curve $c(\mathcal{X}, e_{ij})$ into a probability distribution $P(\mathcal{X}, e_{ij})$ by dividing the curve values $\sum_{t=0}^{T} c(t \mid \mathcal{X}, e_{ij})$. We then define:

$$D(c(\mathcal{X}, e_{ij}), c^\star) = E_{P(t|\mathcal{X}, e_{ij})}[|c(t|\mathcal{X}, e_{ij}) - c^*(t)|], \quad (11)$$

Moreover, optimizing $X_0$ directly in the ambient space can lead to undesired distortions, as there is no guarantee that the edited image remains on the underlying manifold. Instead, based on Equation 4, we select the corresponding $X_t$ from the noisy sequence $\mathcal{X}$ associated with $X_0$ as our optimization variable, where $t$ is a hyperparameter. Because $X_t$ resides in a diffused distribution, it is more fault-tolerant to local edits. Finally, to alleviate computational burden, we adopt a stochastic optimization strategy by sampling exactly one pixel and one timestep per iteration. Specifically, we uniformly choose a pixel $e_{ij}$ from $\mathcal{A}$ in each iteration and randomly choose a pixel $e_k$ from $\mathcal{A}$ at the beginning of the optimization process, keeping $e_k$ fixed throughout. The curve $c(\mathcal{X}, e_k)$ is then normalized to a probability distribution $P(\mathcal{X}, e_k)$, from which we sample a timestep $t_s$ in each iteration.

We update $X_t$ using the gradient descent step $X_t \leftarrow X_t - \eta \nabla_{X_t} |c(t_s|\mathcal{X}, e_{ij}) - c^\star(t_s)|$ with $\eta$ being the learning rate of the SGD optimizer. After optimization, we recover $\bar{X}_0$ from the optimized $\bar{X}_t$ by Equation 4. The optimization algorithm can be applied to various image manipulation tasks by specifying the reference generation curve and adding additional constraints to the objective function.

**Localized Modification**   A typical requirement in many applications is to edit only the image content within a given patch $\mathcal{A}$. However, our standard SGD scheme often modifies the entire image, which is undesirable. Simply applying a mask to $\bar{X}_t$ after each iteration does not solve the problem either, because the mask is applied at the noised level, while the corresponding noiseless image $\bar{X}_0$ can still contain modifications outside $\mathcal{A}$. Instead, we propose to iteratively blend the noised original image, $X_t$, and the noised optimized image, $\bar{X}_t$, via $\bar{X}_t \leftarrow \bar{X}_t \odot \mathcal{A} + X_t \odot \bar{\mathcal{A}}$, where $\bar{\mathcal{A}}$ is the complement of $\mathcal{A}$ and $\odot$ is the pixel-wise product. We perform such blending every 70 iterations for $t > t_{\text{blend}}$. Blending at a sufficiently high noise level ($t > t_{\text{blend}}$) ensures that the noiseless image has seamless patch boundaries around $\mathcal{A}$ by nature of the reverse generation process.

## 5. Image Manipulation Applications

Many image manipulation tasks can be considered as the transformation of visual properties. This section demonstrates how our curve matching algorithm can flexibly perform various image manipulations using only a pre-trained, unconditional diffusion model for image generation. By leveraging our generation curve, we provide a unified framework for these tasks, removing the need for separate supervised models or specialized datasets. Application details and quantitative evaluation are presented in Appendix D.

### 5.1. Semantic Transfer

The semantic transfer task modifies a source region to match the semantic properties (e.g., color, material, texture) of a reference while maintaining other properties (e.g., depth, shape) unchanged. Given the coupling of visual properties within the curve, this problem can be inherently dealt with our curve matching algorithm by specifying the reference curve of a pixel with desired semantic properties. To avoid meaningless transfers from arbitrary references, we choose the reference pixel from a region adjacent to the source area, limiting their difference within expected properties.

Figure 6 shows our semantic transfer results as well as the generation rate curves before and after the optimization. For the first row, the objective is to induce hair growth on the forehead. We select the reference pixel from an adjacent region that not only exhibits the characteristics of hair but is also similar to the source region in other aspects, e.g., both being located on the crown of the head. After optimization, the source curves align with the specified reference, and hair gradually emerges on the forehead.

Since our optimization aims to align each pixel's curve in the editing area with the reference curve, in the figures, we randomly select one pixel from that region to illustrate its curve before and after editing. We do not expect the optimized curves to perfectly align with the reference, because the edited area should maintain harmony with the rest of the image. Therefore, we apply the localized modification technique (see Section 4.2) and stop the iterative optimization algorithm after a predetermined number of steps, before it completely converges to the reference.

### 5.2. Object Removal

Object removal involves replacing an object with the background it obscured, while keeping the rest of the image unchanged. Our curve matching algorithm addresses this by transferring the visual properties of the background to the pixels of the object to be removed. This process shares the same pipeline as semantic transfer, with the reference pixel selected from the expected background.

Figure 7 compares the object removal results of our method and two recent approaches, i.e., SD-XL inpainting (Podell et al., 2023) and an instruction-based method, called ZONE (Li et al., 2023). The shortcomings of these two types of methods are primarily in the following aspects: Inpainting methods that are trained on masked images can be considered a form of re-sampling from the true image distribution conditioned on the unmasked region (Rout et al., 2023). However, the procedure of inpainting often lacks clear guidance to specify the background content after removal, thereby the content in the editing area is unstable, occasionally filled by another object. For instruction-based methods, they achieve image editing through a pre-trained model that accepts textual instructions. However, they sometimes fail to identify the objects by solely relying on text descriptions. In the case of removal, they often fail when dealing with complex occlusion scenarios.

### 5.3. Saliency Manipulation

Saliency manipulation involves altering the saliency of an object while maintaining its identity. The high correlation between visual saliency and the fluctuation of the generation curve allows us to adjust saliency by modifying these fluctuations during curve optimization. Directly specifying a reference curve pattern with only a different fluctuation is challenging. Moreover, we observe that salient object's curve tends to have greater fluctuation and peak with higher values (Figure 2) compared to the non-salient curve which maintains a lower and fixed value. Therefore, to increase (resp. decrease) saliency, we simply maximize (minimize) generation rates, where the reference curve is implicitly specified. We also use the feature alignment loss from UNet as described in (Mou et al., 2023) to preserve image content.

We present the saliency editing results in Figure 8, comparing our results and those of a recent approach RSG (Miangoleh et al., 2023). The results indicate that using curves as a measure and quantification of saliency is reasonable.

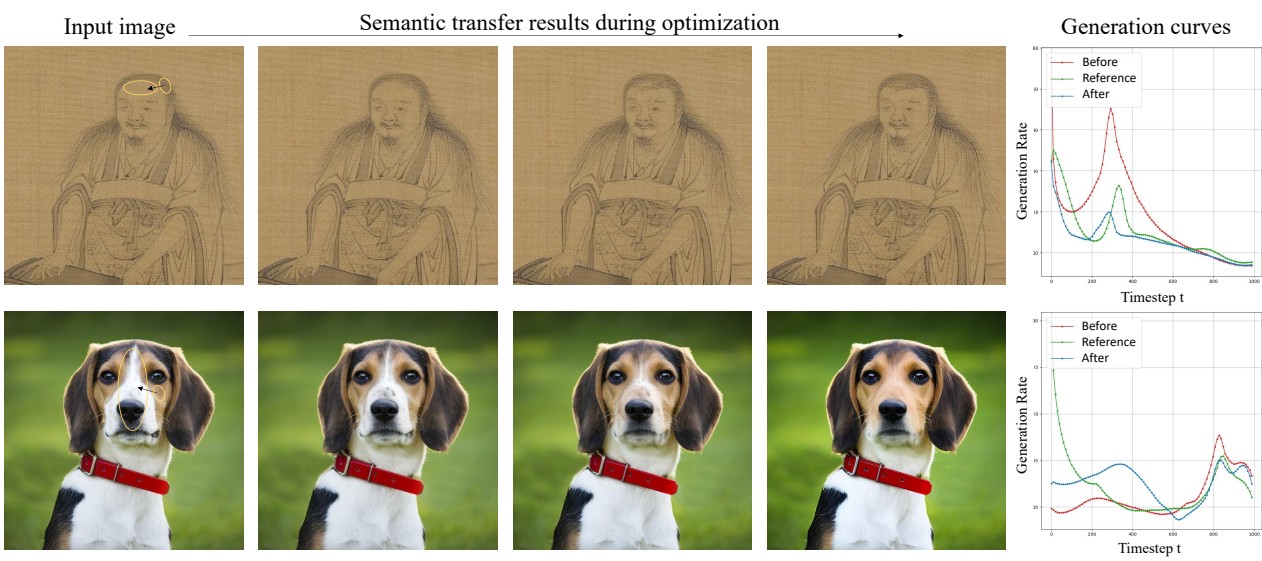

Figure 6. Semantic transfer results. Left: The input image and the transfer results captured at different optimization iterations. The beginning and end of the arrow represent the reference point and the editing region, respectively. Right: Generation curves before and after optimization, as well as the reference curve.

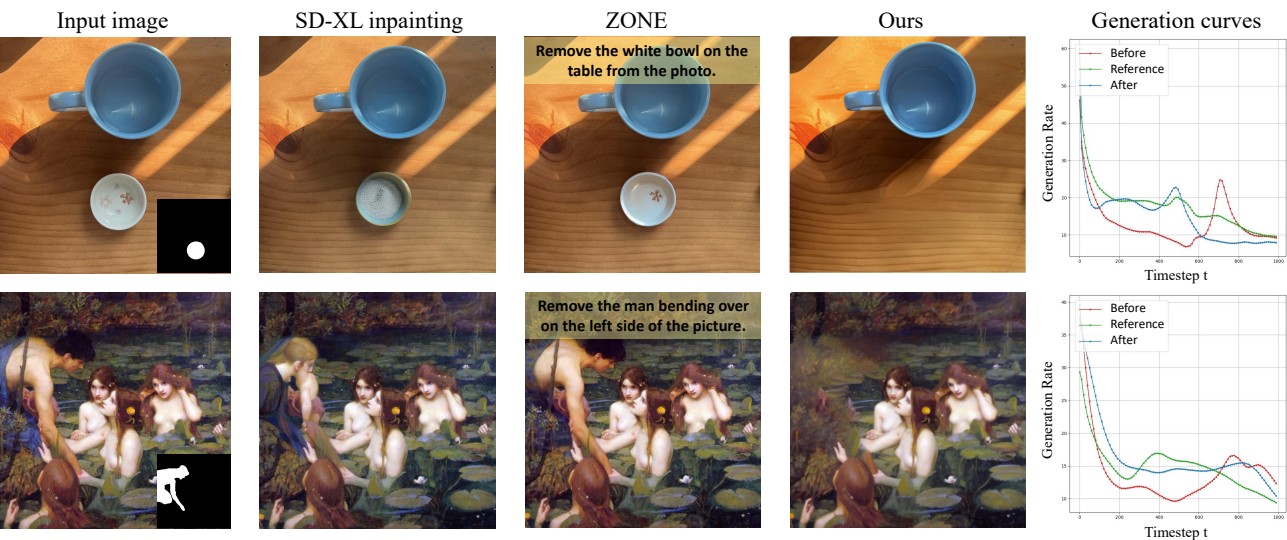

Figure 7. Object removal comparison. From left to right: the input image and object mask, the results of alternative approaches (Podell et al., 2023; Li et al., 2023) and ours, corresponding generation curves during optimization.

By contrast, while existing methods that learn from eye-tracking data can effectively manipulate saliency by end-to-end networks, they often focus on patterns such as color contrast and brightness, thus failing to produce harmonized results across diverse natural images.

### 5.4. Image Blending

The image blending task aims at blending a foreground image with a background image at boundaries. To accomplish this task, given a composite image, we define the boundary between the foreground and background as the salient region, since it contains undesirable and eye-catching seams.

Then we propose to reduce the visual saliency of the boundary region to make a natural transition. With our curve matching algorithm, we follow the same procedure as our saliency manipulation application, but minimize the visual saliency only at the boundary region.

In Figure 9, we show the results of ours and an existing approach (Wu et al., 2019). In our experiments, we found that existing methods often perform well on specific image types and fail to produce satisfactory results across various natural images. Instead, our approach consistently produces visually pleasing boundaries for composite images.

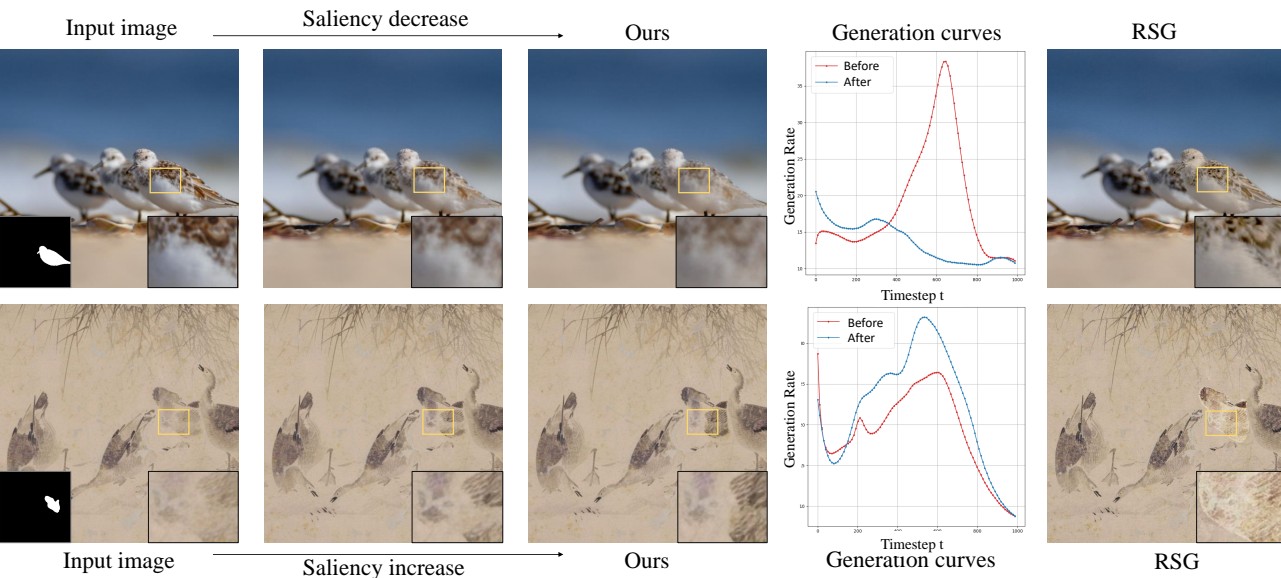

*Figure 8.* Saliency manipulation comparison. From left to right: input images and masks, intermediate and final results of ours, generation curves during optimization, results of an alternative approach (Miangoleh et al., 2023).

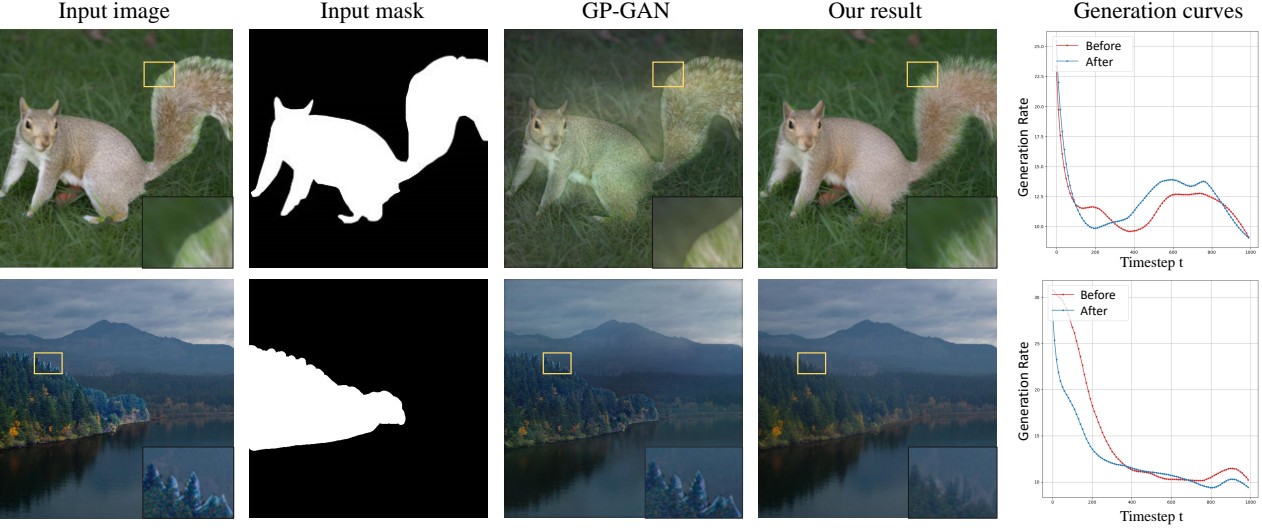

*Figure 9.* Image blending comparison. From left to right: input image, input mask, results of GP-GAN (Wu et al., 2019), our results, and the corresponding generation curves.

## 6. Conclusion

In this work, we propose the generation rate, which corresponds to the local geometric scaling of the manifold over time around an image component. Through comprehensive analytical evaluations, we show that this time-varying geometric deformation exhibits a high correlation with the visual saliency of the image component. In addition, manipulating the generation curves with different loss functions provides a unified framework for a row of image manipulation tasks. Future research could investigate different descriptions of geometric deformation in of data manifold, explore more applications and address the limitations of our generation curve. For instance, our curve optimization algorithm requires first-order differential computation and thus requires approximately 10 minutes for 300 iterations including the pre-processing. On the other hand, for image manipulation tasks, since different objects have varying visual appearances and thus different generation curves, it causes varying convergence speeds and thus different numbers of iterations for curve optimization.

## Impact Statement

This paper presents work whose goal is to advance the field of Machine Learning. There are many potential societal consequences of our work, none which we feel must be specifically highlighted here.

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

## A. Background on Differentiable Manifold

A differential manifold $M$ of dimension $m$ is a topological space such that every point $x \in M$ has a neighborhood homeomorphic to an open set in $\mathbb{R}^m$, and the transition between these local coordinate patches are differentiable. When data are assumed to lie on a low-dimensional manifold $M$ embedded in a higher-dimensional Euclidean space $\mathbb{R}^n$ with $n \geq m$, this manifold structure provides a geometric framework for understanding local neighborhoods, directions, and curvature within the data.

At each point $x \in M$, there is an associated tangent space $T_x M$, which is an $m$-dimensional vector space that linearly approximates the manifold near $x$. Intuitively, $T_x M$ captures the possible "directions" one can move from $x$ while remaining on the manifold. A differential map or forward differential of a smooth function $f : M \to N$ between two manifolds $M$ and $N$, denoted by

$$D_x f : T_x M \to T_{f(x)} N,$$

is a linear map that describes how tangent vectors at $x$ are mapped to tangent vectors at $f(x)$. This allows us to see how local shapes, distances, and densities are distorted when applying the map $f$.

## B. Curve Matching Algorithm

As described in Section 4, our curve matching algorithm manipulates the original image $X_0$ to align the generation curves of a source area $\mathcal{A}$ and a reference pixel $p^\star$. When applied to different image manipulation tasks, we flexibly select the reference pixel and modify the loss function $\mathcal{L}$ to achieve different goals. The default hyperparameter setting is $t = 700$ for transforming $X_0$ to $X_t$ and learning rate $\eta = 0.02$. The iteration number ranges from 30 to 300 for different tasks, and we show their intermediate results during the optimization process. We use the pre-trained unconditional diffusion model, stable-diffusion-2-1-base, for all our experiments.

We present the details in Algorithm 1. The basic idea is to update $X_t$, transformed from the input $X_0$, in order to align the generation curves of the source area $c(\mathcal{X}(X_t), \mathcal{A})$ and the curve $c^\star = c(\mathcal{X}(X_t), p^\star)$ of the reference pixel $p^\star$ for each channel. Specifically, we first transform $X_0$ to $X_t$ via the deterministic diffusion process, and pre-compute the generation curve of the reference pixel $c^\star = r_t(\mathcal{X}(X_t), v^\star)$ and that of a random pixel within the source area $c_a = r_t(\mathcal{X}(X_t), v_a)$. Then we start the iterative optimization. In each iteration, we sample a pixel $p_k$ within the source area randomly and a time $t_k$ based on the cumulative distribution of $c_a$. The optimization variable $X_t$ is transformed to $X_{t_k}$, which is used to compute the generation rate $r_{t_k}(X_{t_k}, v_k)$ and the loss function $\mathcal{L} = |r_{t_k}(X_{t_k}, v_k) - r_{t_k}^\star|$. To avoid back-propagation through the differential equation-based diffusion process, we compute the forward transformation $X_t \to X_{t_k}$ in a non-grad context. Concretely, we first generate the difference $\Delta X = X_{t_k} - X_t$ without tracking any gradients. We then treat $\Delta X$ as a noise term and apply it directly to $X_t$. Then we update $X_t$ with an Adam optimizer. After finishing the optimization, e.g. reaching the maximum iteration number, we recover $X_t$ back to $\bar{X}_0$ via Equation 4.

**Curve updating.** The optimization objective requires to sample $t_k$ according to the curve of the source area, which in turn varies during the optimization and is time-consuming to compute. Firstly, to simplify the computation, we sample a pixel $p_a$ within the source area and compute its generation curve, i.e. $c_a$, to represent the curve of the source area. On the other hand, we update the curve $c_a$ after every $m = 50$ iteration steps. Additionally, for spatial cases when the reference curve also varies, e.g. when the reference pixel lies within the source area, we update the reference curve as well.

## C. Visual Analysis Experiments

### C.1. Visual Saliency Experiment

**Experiment setting.** We validate the connection between the fluctuation of our generation curve and the visual saliency of images, as described in Section 3.2. The experiment is conducted on the MIT saliency benchmark CAT2000 (Borji & Itti, 2015), which provides the collected eye-tracking data of images from human observers and the pre-processed saliency map. For each image, we randomly select one pixel within maximum saliency values and one within minimum values as the salient pixel and non-salient pixel. For the generation curves at the two pixels, we compute its local variance for $t > 200$ (to ignore the abrupt rise when $t$ is close to 0 ) over a sliding window of length $k = 5$ and take their average to represent the curve fluctuation.

**Discussion.** Figure 10 presents the example images and the generation curves corresponding to the selected pixels. For the

**Algorithm 1** Curve Matching Algorithm

**Input:** Image $X_0$, source pixel set $\mathcal{A}$, reference pixel $p^\star$, hyperparameter $t$, iteration number $N$

    /* Initialization */
1: Transform $X_0$ through Eq. 4 to obtain $X_t$ // Initialize the optimization variable
2: Initialize pixel vector $v^\star$ for $p^\star$ and $v_a$ for a randomly sampled source pixel $p_a \in \mathcal{A}$

    /* Pre-processing */
3: $c_a \leftarrow r_t(X_t, v_a)$, $c^\star \leftarrow r_t(X_t, v^\star)$ // Compute curves $c_a$ and $c_r$ for $p_a$ and $p_r$ respectively

    /* Curve optimization with $X_t$ as the variable */
4: **for** $k = 1$ to $N$ **do**
5:     Sample a random source pixel $p_k \in \mathcal{A}$ and initialize its corresponding vector $v_k$
6:     Sample a time $t_k$ based on the normalized $c_a$
7:     Transform $X_t$ to $X_{t_k}$ through Eq. 4 without gradients
8:     Compute generation rate $r_{t_k}(X_{t_k}, v_k)$
9:     Compute the loss $\mathcal{L} = |r_{t_k}(X_{t_k}, v_k) - c^\star(t_k)|$
10:    Update $X_t \leftarrow X_t - \eta \nabla \mathcal{L}$
11: **end for**

    /* Generate the manipulated image from optimized $X_t$ */
12: Transform the edited $X_t$ to $\bar{X}_0$ through Eq. 4

example on the left, the salient pixel corresponds to the blue curve with obviously higher fluctuation than the red curve. In our experiment, we found that our estimated curve fluctuation often reflects the visual saliency well for natural images. For 86% of the images in test set, higher visual saliency leads to higher fluctuation, validating the high consistency between curve fluctuation and visual saliency. However, for some special cases, the noise inherent in eye-tracking data causes the inaccurate spatial location of the salient and non-salient pixels, and thus significantly interfering with our pixel-level calculations. For the line drawing images on the right, our curve fluctuation is not consistent with the ground-truth visual saliency. The red pixel is marked as a non-salient pixel, while it corresponds to a higher curve fluctuation since it is located in the region with dense line drawings.

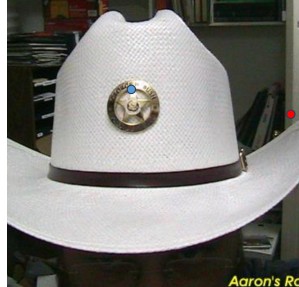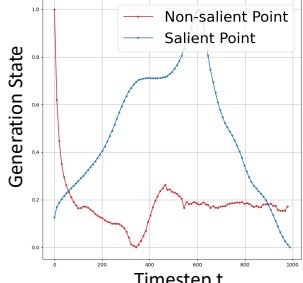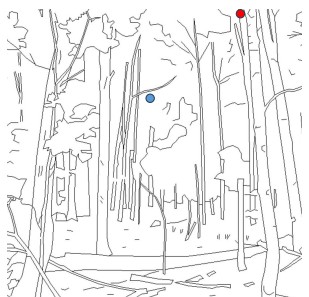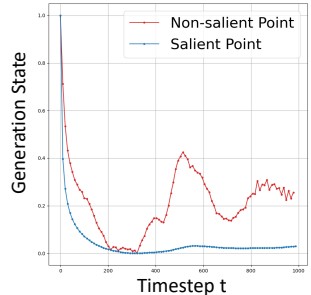

*Figure 10.* The salient pixel (blue) and non-salient pixel (red) on the images and their corresponding generation curves. The left shows that our curve fluctuation is often consistent with the ground-truth saliency for natural images. The right is an inconsistent case with the special line drawing images.

## C.2. Two types of Generation Curves

**Experiment setting.** Given an image and the mask of a small object, we compare our generation curve and another alternative approach, both of which estimate the generation rates of the small object at different timesteps. The alternative approach computes a perceptual-based curve that utilizes perceptual loss to estimate the generation rates of a local region. The experiments are as follows.

- **Perceptual-based curve.** For each timestep during the diffusion process, given the noised image $X_t$, we can predict the $\hat{X}_0$ with DDIM (Equation 4) and decode the corresponding RGB image $\hat{I}$. Then we use the pre-trained DINO

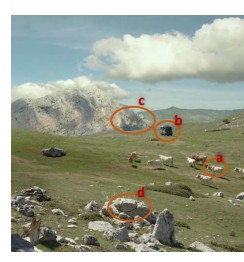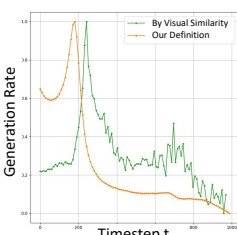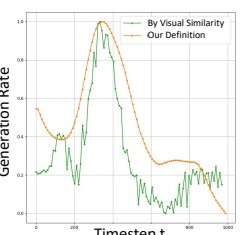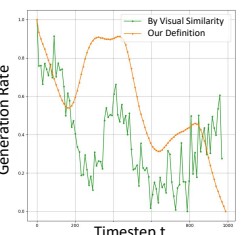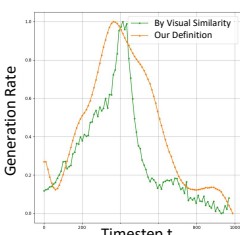

*Figure 11.* More results of the comparison between the perceptual-based curve and our generation curve.

model (Caron et al., 2021) to compute the perceptual loss between the predicted $\hat{I}$ and the original image $I$ w.r.t. the region (mask) $M$ of the specified small object. The perceptual loss is defined as the cosine distance between the features $DINO(\hat{I} \odot M)$ and $DINO(I \odot M)$, corresponding to the generation state at timestep $t$ that ranges from 0 to 1. And its derivatives can be considered as an approximation of the generation rates.

- **Our generation curve.** As described in Section 3.2, we define the generation rate of an object—comprising multiple pixels $\{e_{ij}\}$—as the average of the generation rates of its individual pixels. Given a noised image $X_t$ and a specified mask $M$, we calculate the generation rate $r_t(X_t, v(e_{ij}))$ for each masked pixel vector $v_i$ and compute their average, which represents the generation rate of the object.

**Discussion.** Figure 12 presents the curves of two example images, both ours and the perceptual-based curves. Note that we normalize each curve into the range $[0, 1]$ for better visualization. As described in Section 3.2, the two types of curves exhibit similar trends, especially with their main peaks occurring at close timesteps. It validates that both the two curves reflect the generation rates. On the other hand, compared to the intuitive perceptual-based curves, our generation curve is more applicable to many potential tasks. Firstly, the perceptual loss is object-wise rather than pixel-wise. It prevents its analysis and application to more fine-grained generation patterns. Secondly, the perceptual loss can only distinguish the prominent foreground features. Consequently, the perceptual-based curve often exhibits heavy noise or even negative values with less prominent visual features, especially the flat background, as shown in Figure 13.

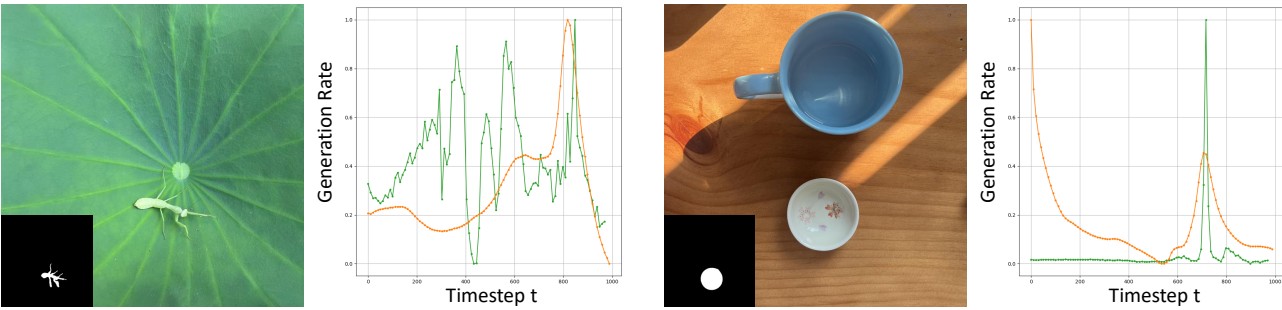

*Figure 12.* Two types of generation curves for the masked small object in the image. Green: the perceptual-based curve; Orange: our generation curve. All the curves are normalized into range $[0, 1]$.

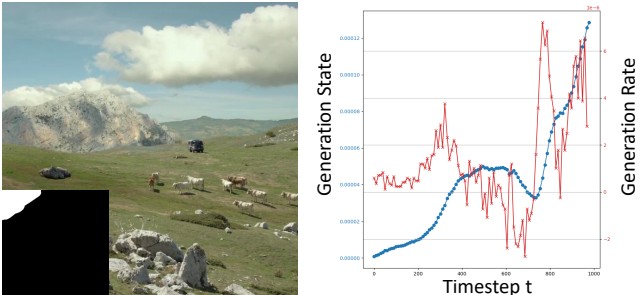

*Figure 13.* We plot the generation state (blue) and generation rate (red) estimated by the perceptual-based loss. For background areas without prominent visual features, the perceptual-based curves (red) tend to exhibit heavy noise or even negative generation rates.

## D. Application Details

We discuss more application details in this section. We conduct quantitative experiments utilizing Vision-Language models (Radford et al., 2021; Caron et al., 2021) and user studies as discussed in D.1, with results shown in Table 1.

*Table 1.* Quantitative evaluations of object removal, saliency editing (increasing/decreasing), image blending, semantic transfer.

| Object Removal | CLIP$_{\text{dir}}\uparrow$ | CLIP$_{\text{sim}}\downarrow$ | DINO$_{\text{sim}}\downarrow$ | Effectiveness$\uparrow$ | Realism$\uparrow$ |
|---|---|---|---|---|---|
| ZONE (Li et al., 2023) | 0.2589 | 0.4824 | 0.433 | 4.48 | 6.3 |
| SD-XL inpainting (Podell et al., 2023) | 0.2617 | 0.4787 | 0.422 | 5.63 | **7.52** |
| Ours | **0.2629** | **0.4782** | **0.416** | **6.01** | 6.62 |
| **Saliency Editing** | CLIP$_{\text{dir}}\uparrow$ | CLIP$_{\text{sim}}\downarrow$ | DINO$_{\text{sim}}\downarrow$ | Effectiveness$\uparrow$ | Realism$\uparrow$ |
| RSG (Miangoleh et al., 2023) | 0.2518/0.2508 | **0.4903**/0.4890 | 0.4745/0.4763 | **6.76/6.95** | 8.14/7.52 |
| Ours | **0.2521/0.2516** | 0.4908/**0.4876** | **0.4740/0.4653** | 6.32/6.77 | **8.58/8.53** |
| **Image Blending** | CLIP$_{\text{dir}}\uparrow$ | Effectiveness$\uparrow$ | Realism$\uparrow$ | **Semantic Transfer** | CLIP$_{\text{diff}}$ |
| GP-GAN (Wu et al., 2019) | 0.3221 | 5.68 | 4.86 | Ours | 0.0108 |
| Ours | **0.3273** | **8.11** | **7.75** | / | / |

### D.1. Quantitative evaluation setup

The evaluation of these tasks often lacks standard metrics and datasets. Therefore, we take a general setting using common Vision-Language models (Radford et al., 2021; Caron et al., 2021) and user studies, which are also adopted in (Sheynin et al., 2023; Miangoleh et al., 2023).

**Similarity metrics.**

- We use CLIP$_{\text{dir}}$ (Sheynin et al., 2023) to reflect whether the image editing direction is consistent with that indicated by a text prompt. This metric is defined as

$$\text{CLIP}_{\text{dir}} = \frac{\langle e_{\text{cap}}(T),\, e_{\text{img}}(I_a) - e_{\text{img}}(I_b)\rangle}{\|e_{\text{cap}}(T)\|\, \|e_{\text{img}}(I_a) - e_{\text{img}}(I_b)\|}, \tag{12}$$

where $e_{\text{cap}}(\cdot)$ and $e_{\text{img}}(\cdot)$ are the CLIP encoders for text and image, respectively. $T$ is the embedding of the text prompt that describes the editing direction. $I_b$ and $I_a$ are the images before and after editing. A *higher* CLIP$_{\text{dir}}$ indicates a more effective editing. We use instructional descriptions such as *"decrease visual attention"* for saliency editing, *"remove the car from the picture"* for object removal, and *"make the boundary transition smoother"* for image blending.

- We use CLIP$_{\text{sim}}$ and DINO$_{\text{sim}}$ to measure the similarity between images before and after editing. They are defined as

$$\text{CLIP}_{\text{sim}}(I_1, I_2) = \frac{\langle e_{\text{img}}(I_1),\, e_{\text{img}}(I_2)\rangle}{\|e_{\text{img}}(I_1)\|\, \|e_{\text{img}}(I_2)\|}, \tag{13}$$

$$\text{DINO}_{\text{sim}}(I_1, I_2) = \frac{\langle d_{\text{img}}(I_1),\, d_{\text{img}}(I_2)\rangle}{\|d_{\text{img}}(I_1)\|\, \|d_{\text{img}}(I_2)\|}, \tag{14}$$

where $e_{\text{img}}(\cdot)$ is the CLIP image encoder and $d_{\text{img}}(\cdot)$ is the DINO encoder. We evaluate CLIP$_{\text{sim}}$ and DINO$_{\text{sim}}$ specifically over the edited region. These metrics are used to evaluate the object removal and saliency editing tasks. *Lower* values indicate larger change (thus more effective editing).

- We use CLIP$_{\text{diff}}$ to examine whether the edited images match better with the text prompt describing the desired edit than before. It is defined as

$$\text{CLIP}_{\text{diff}} = \text{CLIP}(T,\, I_a) - \text{CLIP}(T,\, I_b), \tag{15}$$

where $T$ is a natural-language description of the desired editing, and $I_a$, $I_b$ are the images *after* and *before* the edit. CLIP is the text-image similarity score

$$\text{CLIP}(T, I) = \frac{\langle e_{\text{cap}}(T),\, e_{\text{img}}(I)\rangle}{\|e_{\text{cap}}(T)\|\, \|e_{\text{img}}(I)\|}. \tag{16}$$

This metric is used to evaluate the semantic transfer task. A *positive* CLIP$_{\text{diff}}$ means $I_a$ aligns more closely with the target description $T$ than $I_b$, indicating a successful semantic transfer.

**User study.** We present the results independently to 8 participants and ask them to rate the effectiveness and realism respectively from 1 to 10. The study uses 20 test examples for image blending, 15 for object removal, 30/30 for saliency increasing/decreasing.

### D.2. Semantic Transfer

The semantic transfer application shares the same pipeline with our object removal application, except that we select a representative pixel of a surrounding object as the reference. We demonstrate more visual results in Figure 14. It enables interactive editing with only a click on the image to specify the reference. However, since the generation curves involve various visual properties, sometimes the transfer results don't exhibit the desired visual effects. For example, as shown in Figure 15, it transfers the material but not the color in the two examples.

As shown in Table 1, the positive value of CLIP$_{\text{diff}}$ indicates that the images are edited towards the semantic transfer and evaluate the effectiveness of our method.

### D.3. Object Removal

**Experiment setting.** We conduct the comparison experiment on the test set from Emu Edit benchmark (Sheynin et al., 2023). For the object removal task, it provides the original images, the input and output captions, and the text instructions to specify the objects to be removed. We pre-process the test set with an image segmentation tool to obtain the masks of the objects to be removed. The quantitative evaluation is performed on a random subset of 100 images from this test set. Note that there's no need for the training set since we utilize the pre-trained models in all our experiments.

The related works often utilize image inpainting (Criminisi et al., 2004; Suvorov et al., 2022; Lugmayr et al., 2022; Podell et al., 2023) or instruction-based image editing (Brooks et al., 2023; Pan et al., 2023; Hertz et al., 2023) for the object removal task. The former resamples the editing area, which is then likely to fall into the high-density distributions, i.e. image background. The latter fine-tunes the pre-trained image generation model to take the text instructions as conditions. We compare with the SD-XL inpainting (Podell et al., 2023) and instruction-based method ZONE (Li et al., 2023) in our experiments:

- **SD-XL inpainting.** We input the original image and the object mask to the SD-XL model, i.e. stable-diffusion-xl-1.0-inpainting-0.1, and obtain the inpainted image as the result.

- **ZONE.** We take the original image and the text instruction as input and the generated image as result. Note that the pre-processed object mask is used in the composition of the generated image for a precise local editing.

- **Ours.** The input includes the original image, the object mask as the source area, a selected surrounding background pixel as the reference. Note that the tedious selection of reference pixels can be omitted for simple cases with a uniform background, as described in the following implementation.

**Implementation.** To alleviate the manual selection of reference pixels, we adopt an approximation solution for the simple cases. That is, we divide the test set into two types: one containing the target objects located on uniform backgrounds, and the other with complex and varying backgrounds. For the former, instead of using the generation curve at a manually selected reference pixel, we define a fixed pseudo curve to replace it. Specifically, since the generation curves at background locations often follow the common pattern with stable and lower values, we directly minimize the generation rate values at the target area, i.e. $\mathcal{L} = |r(X_{t_i}, p_i)|$. For the latter, we manually select the reference pixels and invoke our curve matching algorithm as described in Algorithm 1. We experimentally found that the pseudo curve solution is enough for most images (more than 80 out of the 100 images in our quantitative evaluation, Table **??** in the main paper).

**Results and failure cases.** We present more results of object removal in Figure 16. Table 1 validates that our approach performs better in terms of deleting the objects, obtaining the best effectiveness yet slightly lower realism than SD-XL. However, we also notice some failure cases when the background is complex. As shown in Figure 17, sometimes it may be hard to specify a reference pixel representative for the background. This often causes distorted results at the region of the object to be removed.

### D.4. Saliency Manipulation

**Experiment setting.** Saliency manipulation refers to increasing or decreasing the saliency of a specific object as one expects, while maintaining the image content as unchanged as possible (Aberman et al., 2021; Mejjati et al., 2020; Jiang et al., 2021; Miangoleh et al., 2023). The input contains the original image and a mask indicating the region to be edited. In our experiments, we compare against a recent saliency-based image manipulation approach (Miangoleh et al., 2023). This approach, denoted as RSG, optimizes the image with a saliency loss using a pre-trained saliency model and a realism loss to prevent frequent unrealistic edits. We use their released code and the parameters in our experiments. As for our approach, we eliminate the reference curve, but directly minimize or maximize the generation rate values at the editing region to control the saliency. At the same time, we define a feature alignment loss to preserve the original image content. The feature alignment loss is defined as the difference between the U-Net intermediate feature maps before and after the editing. In summary, we replace the loss in Algorithm 1 as $\mathcal{L} = \lambda_1 |r_{t_k}(X_{t_k}, p_k)|^{\lambda_2} + \sum_i |U_i(X_{t_k}) - U_i(X_t)|$, where $\lambda_1 = 50$ is a weighting parameter and $\lambda 2$ takes values 1 or $-1$ to decrease or increase the saliency, respectively. And $U_i$ represents the U-Net layers which outputs the intermediate feature maps. We set the iteration number $N = 70$ for the saliency manipulation task.

**Results and failure cases.** We present the saliency increase and decrease examples in Figure 18 and 19. As the iteration number increases during the optimization, our approach obtains edited objects with saliency varies as expected. Increasing the saliency results in sharp color contrast and more visual details, while decreasing the saliency results in faded visual effects. The quantitative results in Table 1 validate that we achieve a balance between effectiveness and realism, while RSG (Miangoleh et al., 2023) produces prominent but unrealistic image regions. It is worth noting that all the saliency manipulation approaches require a balance between local editing and the preservation of image content. Sometimes, forcing the saliency variation may cause the altering of the original identity and object distortion. We also present the failure cases in Figure 20.

### D.5. Image Blending

**Experiment setting.** Image blending aims to create a natural boundary transition for the compositional images (Pérez et al., 2003; Wu et al., 2019; Zhang et al., 2020; 2021; Xing et al., 2022; Niu et al., 2024). We evaluate the image blending results on the iHarmony4 dataset (Cong et al., 2020). It provides the synthesized composite image with inconsistent foreground and background, as well as the corresponding foreground masks. To conduct the experiment with our approach, we apply the erosion filter with kernel size $k = 3$ on the given mask and take the eroded region as the source area of our curve matching algorithm. We use the algorithm with loss $\mathcal{L} = \lambda_1 |r_{t_k}(X_{t_k}, p_k)| + \sum_i |U_i(X_{t_k}) - U_i(X_t)|$ to complete this task, where $\lambda_1 = 50$ is a weighting parameter and $p_i$ are the pixels within the eroded region of the mask. We set the iteration number $N = 100$ for the image blending task.

**Results and failure cases.** When compared to (Wu et al., 2019) in Figure 9, this approach requires both the complete background and foreground images as input, Therefore, we utilize online tools that combine the inpainting (Suvorov et al., 2022) and post-processing techniques to provide a suitable background, as shown in Figure 23. We present more results in Figure 21. Table 1 shows that we obtain obviously better blending results with a natural and smooth transition at the boundary. However, the smooth transition often corresponds to blurred details. As a consequence, the image details might be smoothed out during the blending, such as the beak of the bird and the leaves on the tree in Figure 22.

Input image        Semantic transfer results during optimization

Figure 14. More results of semantic transfer. The left column shows the input image, where the arrows indicate the semantic transfer from the reference to source area. From left to right, we show the intermediate results during the optimization.

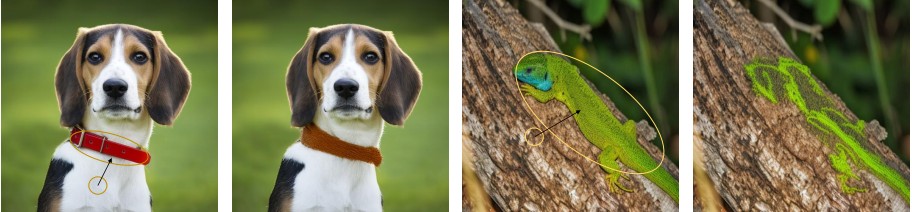

Figure 15. Failure cases of semantic transfer. The material and depth features are transferred but not the color.

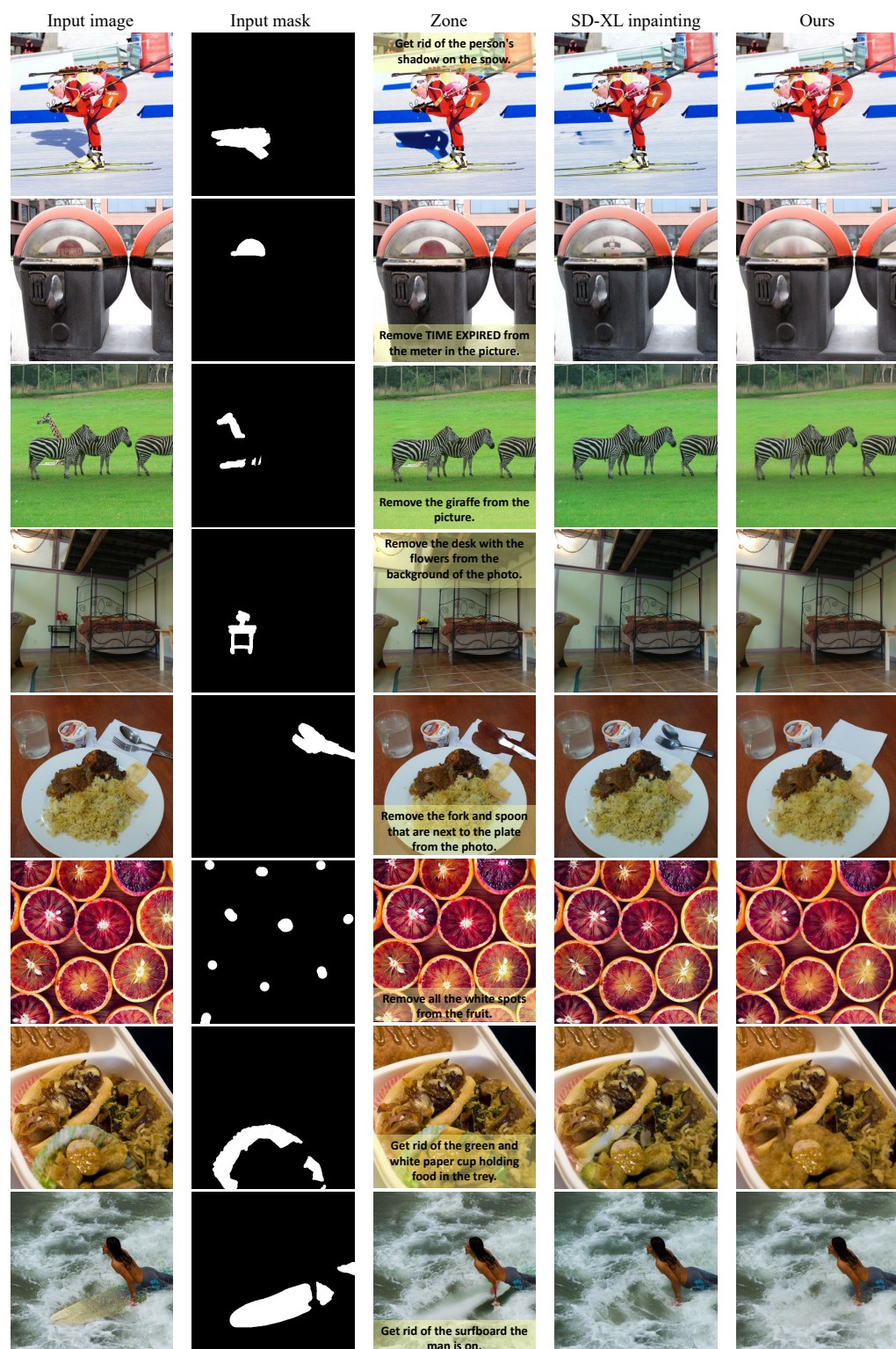

*Figure 16.* More object removal results. From left to right: the input image and the input object mask, the results of the instruction-based method ZONE (Li et al., 2023), SD-XL inpainting (Podell et al., 2023), and our approach.

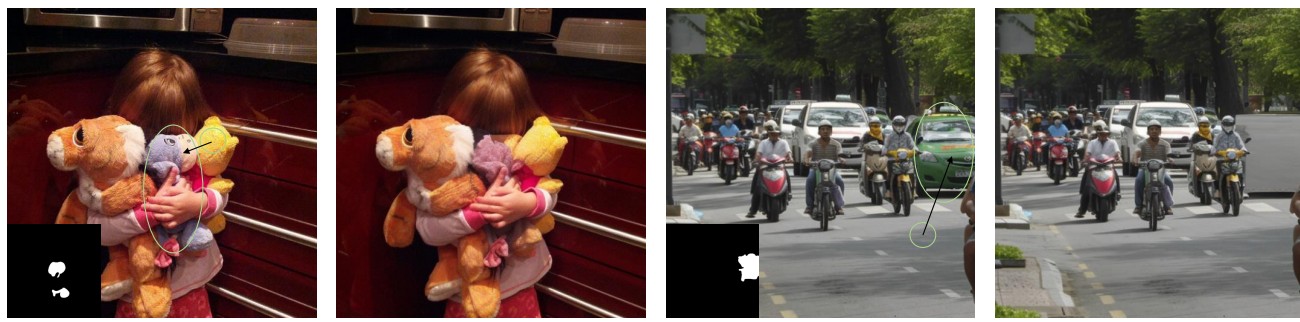

*Figure 17.* Failure cases of object removal application.

Input image      Saliency increase        Ours        RSG

*Figure 18.* Saliency increasing results. From left to right: the input image, the intermediate and final results of our approach, the result of RSG (Miangoleh et al., 2023).

Input image      Saliency decrease      Ours      RSG

*Figure 19.* Saliency decreasing results. From left to right: the input image, the intermediate and final results of our approach, the result of RSG (Miangoleh et al., 2023).

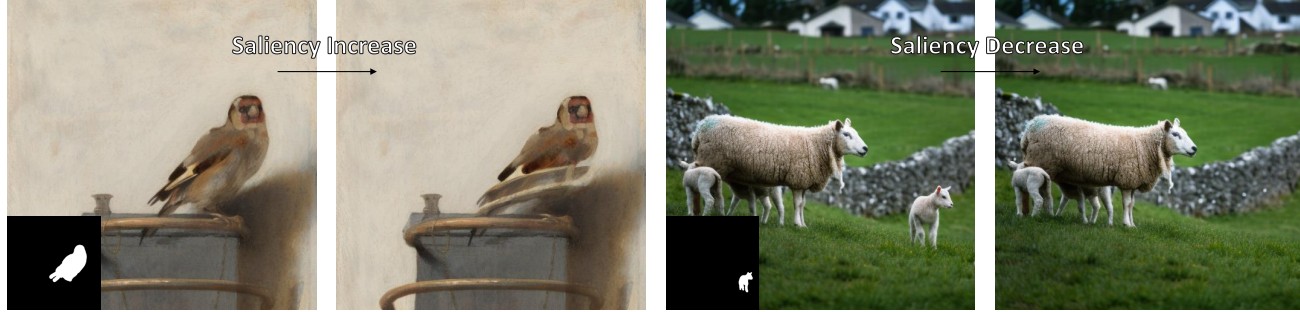

*Figure 20.* Failure cases of saliency manipulation.

| Input Composite Image | Ours | Input Composite Image | Ours |
| --- | --- | --- | --- |

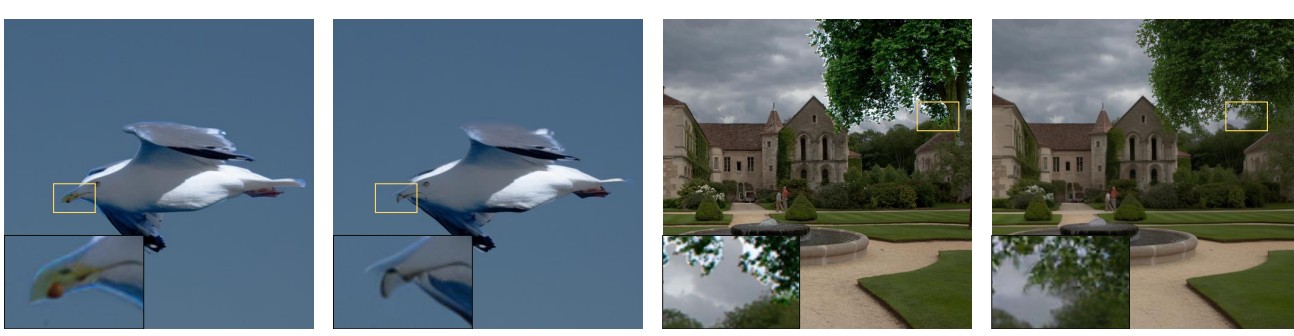

*Figure 21.* Image blending results. For each example, we show the input composite image and the foreground mask, as well as the result of our approach.

*Figure 22.* Failure cases of image blending. Although we achieve a natural blending at the boundary, it loses the small details such as the beak of the bird and the leaves on the tree.

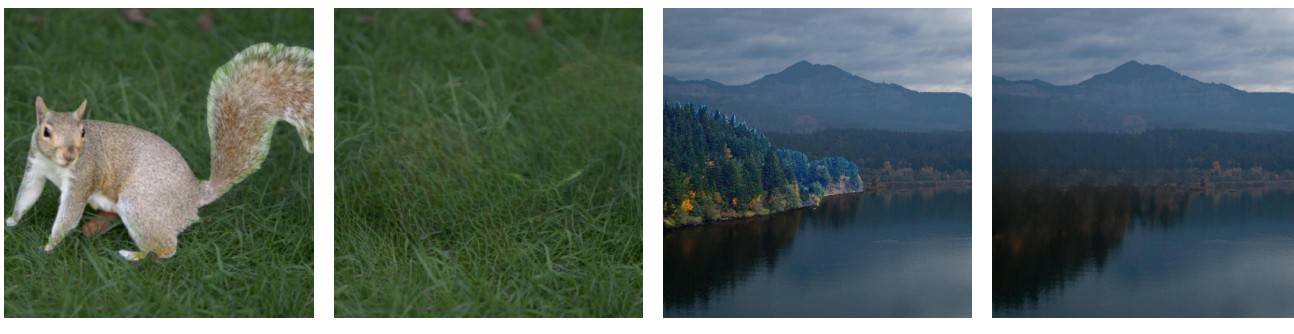

*Figure 23.* Provided background for GP-GAN (Wu et al., 2019) in image blending.

