# OpenReview forum: "Varying Manifolds in Diffusion: From Time-varying Geometries to Visual Saliency"
_ICML.cc/2025/Conference — Submitted to ICML 2025_

### Official Review · Reviewer_Zb6M · 2025-02-17

**Overall Recommendation:** 3

**Summary:**

This paper analyses the variance of pixel intensity over generation timesteps for salient versus non-salient regions of the image. They call the rate of change the generation rate. Specifically, they find that the salient regions of the image generally have a higher variance than non-salient points (86% of the time). They use this finding to perform image manipulation tasks, optimizing for a particular x_t to have the generation rate of a specified region in the image match the generation rate of another specified region of the image. Using these techniques they show qualitative examples of edits they make.

**Claims And Evidence:**

The claims made in the paper are not very well substantiated. On a dataset of 100 images, they show that 86% of the images have higher variance in the salient regions of the images. This single small dataset is not enough to say conclusively that what they are claiming is indeed the case, especially since even on the one dataset, it's more of a general trend than definitively showing that salient regions have higher windowed variance.

Beyond this, they have no quantitative analysis of their work, such as on the edits they propose. The examples provided show that it is possible to make the edits they propose, but without quantitative results, it's difficult to say whether this is consistently the case.

**Essential References Not Discussed:**

I would like to see comparisons in image blending with more recent methods. A quick search leads me to a WACV 2020 paper with code available by Zhang et al. If The authors find more recent work to compare to I would accept that as well.

`L. Zhang, T. Wen, and J. Shi, “Deep Image Blending,” in Proceedings of the IEEE Winter Conference on Applications of Computer Vision (WACV), 2020, pp. 231–240.`

**Experimental Designs Or Analyses:**

See `Methods And Evaluation Criteria`

**Methods And Evaluation Criteria:**

## Visual saliency analysis
I would expect to either see these results on a dataset larger than 100 images or show a set of different datasets.

## Baselines for qualitative results for proposed edits
I am not entirely familiar with the current state of the art in the tasks shown, however I find it concerning that the only method compared do for image blending is from 2019.

## Quantitative results for proposed edits
It seems this is an important section missing in the paper. Without quantitative results on a set of varied and large datasets, it is impossible to know if the proposed edits lead to consistent and generalizable improvements in image editing.

**Other Comments Or Suggestions:**

I do not understand what the five different plots in Figure 5 show. I assume they are for specific regions of an image, but I am unclear about that.

Also, I assume the tasks shown involve image inversion before editing; however, that was never made clear.

**Other Strengths And Weaknesses:**

See `Relation To Broader Scientific Literature `

**Questions For Authors:**

If the authors show improvements in more recent methods with quantitative results on a set of datasets, I would happily raise my rating. Even if the authors can show correlations on larger datasets between the saliency of a region and the windowed variance, this would be enough for me to improve my rating. For the time being, given a lack of quantitative analysis, I am unsure of if the proposed method is indeed responsible for the improvements shown.

**Relation To Broader Scientific Literature:**

A strength of this paper is the originality of what is being proposed. I have not seen an analysis like this, and if it is indeed the case, then I could see it inspiring future work analyzing the correlation between saliency and variance in the generation rate.

**Theoretical Claims:**

The proposed method theoretically makes sense given the tasks at hand. If the variance in the generation rate is indeed correlated with the saliency of the region in the image it is reasonable to see how it would be able to make the proposed edits.

---

> ### Author Rebuttal · Authors · 2025-04-01
>
> Thank you for your detailed review! Below we address the questions.
>
> **Q1. Larger Dataset for Visual Saliency Evaluation.**
>
> We have expanded our visual saliency evaluation to include the full MIT saliency benchmark CAT2000, consisting of 2000 diverse-category images. Our results indicate that 81% of randomly chosen salient points exhibit higher curve variance compared to non-salient points. Furthermore, we conducted rigorous statistical analyses as suggested by Reviewer xDCq on the above data:
>
> - Point-Biserial Correlation: $(r = 0.297, p=1.36e^{-69})$
>
> - Independent Samples t-test: $(t = 18.050, p=1.36e^{-69})$
>
> The statistical tests further confirm a significant positive correlation between generation curve fluctuations and visual saliency, and statistically significant differences between salient and non-salient curves. Visual results are presented in the https://limewire.com/d/3h4XH#SSSDAgZ0rL Figure 2.
>
> **Q2. Quantitative Evaluation on Applications.**
>
> We have included quantitative evaluations for all four proposed applications in Tab.1, Appendix D (lines starting from 648). Following related works, we employ large Vision-Language models such as CLIP, alongside user studies, to quantitatively assess the performance. Results demonstrate our method outperforms or is competitive with current state-of-the-art approaches across evaluation metrics. Detailed discussions of quantitative comparisons can be found in the final paragraph of each application subsection within Appendix D.
>
> **Q3. Comparison with Recent Image Blending Baselines.**
>
> According to our investigation, recent relevant works include [Wu2019, Zhang2020, Zhang2021, Xing2022], summarized in a comprehensive survey [Niu2025](page 8-10). Unfortunately, [Zhang2021] and [Xing2022] do not provide publicly available implementations. Therefore, our comparisons primarily focus on [Wu2019] and [Zhang2020], which were contemporaneously published in 2019.
>
> Upon evaluation, [Zhang2020](Deep Image Blending) frequently introduces undesirable background color leakage into composite objects, leading to noticeable color distortion in natural images. We illustrate these issues with specific examples in https://limewire.com/d/3h4XH#SSSDAgZ0rL Figure 4. Consequently, we select [Wu2019] as the primary baseline due to its consistently better performance in maintaining visual coherence in blended images.
>
> [Wu2019] Huikai Wu, Shuai Zheng, Junge Zhang, and Kaiqi Huang. GP-GAN: Towards realistic high-resolution image blending. ACM MM, 2019.
>
> [Zhang2020] L. Zhang, T. Wen, and J. Shi (2020). Deep Image Blending. WACV, 2020.
>
> [Zhang2021] He Zhang, Jianming Zhang, Federico Perazzi, Zhe Lin, and Vishal M Patel. Deep image compositing. WACV, 2021.
>
> [Xing2022] Yazhou Xing, Yu Li, Xintao Wang, Ye Zhu, and Qifeng Chen. Composite photograph harmonization with complete background cues. ACM MM, 2022.
>
> [Niu2025]Li Niu, Wenyan Cong, Liu Liu, Yan Hong, Bo Zhang, Jing Liang, Liqing Zhang. Making Images Real Again: A Comprehensive Survey on Deep Image Composition. arXiv preprint arXiv:2106.14490(v6), 2025.
>
> **Q4. Clarification of Curves in Figure 5.**
>
> We clarify that the curves presented in Figure 5 are generated at the pixel level, intended to illustrate the approximation of curve shapes discussed in Section 4.1. Since the objective is solely to compare shape approximations rather than interpret specific semantic content, pixel-level representation sufficiently serves this purpose. We will explicitly mention this clarification in the figure caption to avoid potential confusion.
>
> **Q5. Image Inversion for Real Images.**
>
> Yes, image inversion is required and employed for all real-image analyses and subsequent editing tasks discussed. To enhance clarity, we will explicitly mention the diffusion-model inversion at the conclusion of Section 3.1 (around line 134).

---

### Official Review · Reviewer_SNns · 2025-03-11

**Overall Recommendation:** 3

**Summary:**

This work analyzes the correlation between image features relevant to visual saliency and the local deformation of the data manifold induced during the reverse diffusion process, referred to as the generation rate. Empirically, the authors find that the generation curve—the ordered sequence of generation rates computed for each pixel throughout the reverse diffusion process—is strongly associated with visually salient features. The authors propose an image editing technique that exploits the generation curve. The effectiveness of the proposed method is demonstrated across various image manipulation tasks, including semantic transfer, object removal, and image blending.

**Claims And Evidence:**

While the idea of geometrically analyzing data manifold behavior during the reverse diffusion process using differential maps is interesting, I think the experiments lack sufficient baselines to effectively compare and assess the performance of the proposed method. The detailed comments regarding the experiment, particularly comparisons, are listed in “Experimental Designs or Analyses” section.

**Essential References Not Discussed:**

Given the target applications, it would be valuable for the authors to discuss the following work in the revised version:

1. Toward Realistic Image Compositing with Adversarial Learning, Chen *et al.*, CVPR 2019
2. ObjectStitch: Object Compositing with Diffusion Model, Song *et al.*, CVPR 2023
3. Resolution-robust Large Mask Inpainting with Fourier Convolutions, Suvorov *et al.*, WACV 2022
4. SDEdit: Guided Image Synthesis and Editing with Stochastic Differential Equations, Meng *et al.,* ICLR 2022
5. RePaint: Inpainting using Denoising Diffusion Probabilistic Models, Lugmayr *et al.*, CVPR 2022
6. Improving Diffusion Models for Inverse Problems using Manifold Constraints, Chung *et al.*, NeurIPS 2022
7. Diffusion Posterior Sampling for General Noisy Inverse Problems, Chung *et al.*, ICLR 2023

**Experimental Designs Or Analyses:**

The experimental section could be further improved by incorporating additional baselines to better evaluate the effectiveness of the proposed technique. Specifically, I am curious why existing training-free methods that leverage pre-trained diffusion models for image editing and inverse problems were not included in the comparison, such as:

1. (Object Removal) SDEdit: Guided Image Synthesis and Editing with Stochastic Differential Equations, Meng *et al.,* ICLR 2022
2. (Object Removal) RePaint: Inpainting using Denoising Diffusion Probabilistic Models, Lugmayr *et al.*, CVPR 2022
3. (Object Removal) Improving Diffusion Models for Inverse Problems using Manifold Constraints, Chung *et al.*, NeurIPS 2022
4. (Object Removal) Diffusion Posterior Sampling for General Noisy Inverse Problems, Chung *et al.*, ICLR 2023
5. (Image Blending) Toward Realistic Image Compositing with Adversarial Learning, Chen *et al.*, CVPR 2019

Additionally, providing more qualitative results that demonstrate the proposed method's superiority over the baselines would make the assessment more concrete.

**Methods And Evaluation Criteria:**

The description of the proposed method is generally clear but requires some clarification (detailed in “Questions for Authors” section).

Regarding the evaluation criteria, the choice of metrics appears appropriate for the considered applications.

**Other Comments Or Suggestions:**

I have no further comments.

**Other Strengths And Weaknesses:**

I have no further comments on the strengths and weaknesses.

**Questions For Authors:**

To summarize, I would like to ask the following questions to the authors:

1. Could you elaborate on how Equation 8 can be derived from Equation 4?
2. In line 211, the text states that perceptual metrics such as LPIPS, computed using the estimated $\hat{X}_0$ and their time derivatives, can be interpreted as the generation rate defined in Equation 5. I find this statement somewhat confusing, as my understanding is that the generation rate quantifies how the generative mapping from a prior distribution (e.g., Gaussian) to a data distribution deforms the data manifold over time. However, metrics like LPIPS are independent of the data distribution.
3. According to Section 4.2, a reference generation curve of an image patch is required for editing the source patch. However, how can this curve be obtained for real images that are not synthesized through the reverse diffusion process? Are techniques such as inversion incorporated to compute the generation rate and curve?
4. Although I may have misunderstood, could you clarify how visual saliency relates to the downstream applications discussed in the experiment section? While there is an observed empirical relationship between salient pixels in images and their rate of change during the reverse diffusion process, is there any insight into how this can be leveraged for image editing? Additionally, I wonder whether this approach can be applied to edit arbitrary regions in the input image that are not inherently salient.
5. Could you discuss the reasons why the works mentioned in the "Experimental Designs or Analyses" section were not considered in the evaluation? How does the proposed method compare to these works in terms of performance?

**Relation To Broader Scientific Literature:**

This paper has the potential to deepen our understanding of diffusion-based generative models through the lens of the manifold hypothesis. The authors' findings can be applied to various image editing tasks, including object removal and image blending.

**Theoretical Claims:**

This paper does not include any theoretical claims or proofs.

---

> ### Author Rebuttal · Authors · 2025-04-01
>
> Thank you for your detailed review! Below we address the questions.
>
> **Q1-1. Expanded Discussion on Baselines - Object Removal.**
>
> Methods for object removal generally fall into two categories: image inpainting and instruction-based editing (as detailed in the recent survey [Huang2025]). We selected effective representatives from both ([Podell2024, Li2024]).
>
> Regarding the specific baselines:
>
> - **RePaint[Lugmayr2022]**: our SDXL-inpainting baseline [Podell2024] is fundamentally built on Repaint's principles with a more advanced base model (SDXL) and specialized post-training on inpainting datasets. The theoretical justifications (Repaint+[Rout2023]) for Repaint-based inpainting is discussed around line 300.
>
> - **Inverse-Problem-Based Methods[Chuang2022, Chuang2023]**: We quantitatively compared with the more updated [Chuang2023] (see Table 1 and Figure 3 in https://limewire.com/d/3h4XH#SSSDAgZ0rL).  Our method outperforms in editing direction-likely due to guidance from our reference curves-while they demonstrate greater visual changes.
>
> - **SDEdit[Meng2022]:** As an early diffusion-based editing method that incorporates guiding instructions, SDEdit is often regarded as a form of instruction-based editing. Subsequent methods, such as InstructPix2Pix, have reported significantly improvement upon SDEdit. Since our selected SOTA baseline, Zone[Li2024], demonstrated clear superiority over these methods, we consider explicit comparison to SDEdit unnecessary.
>
> **Q1-2. Expanded Discussion on Baselines - Image Blending.**
>
> [Chen2019] focuses on replacing an object with a semantically similar but visually distinct one (closer with image-to-image translation), whereas our method emphasizes seamless pixel-level blending of an inserted object with its surrounding. We discussed an extensive comparison in response to Reviewer Zb6M's Q3, identifying our baseline [Wu2019] as the most relevant and strongest open-source baseline.
>
> In the original paper, additional qualitative results for four tasks are provided in Appendix Figure 14-21.
>
> [Rout2023]Rout et al.. A theoretical justification for image inpainting using denoising diffusion probabilistic models. 2023.
>
> [Huang2025] Huang et al.. Diffusion Model-Based Image Editing: A Survey. TPAMI 2025.
>
> **Q2. Essential References Not Discussed.**
>
> We will integrate the discussion in Q1 and the following to our Appendix D.
>
> - **LaMa[Suvorov2022]** Although effective, the GAN-based LaMa frequently introduces grid-like artifacts when compared to diffusion-based inpainting as also noted in [Lugmayr2022].
> - **ObjectStitch[Song2023]** They leverage guided diffusion models for object compositing, focusing on semantic consistency rather than boundary smoothing. Compared to our method, they provide more diverse outputs at the expense of precise pixel alignment.
>
> **Q3. Derivation of Equation Eq.8 from Eq.4.**
>
> Eq.8 can be explicitly derived from Eq.4 by treating $X_{t-\Delta t}$ as a function $g(X_t)$: $X_{t-\Delta t} =g(X_t) = \sqrt{\frac {\alpha_{t-\Delta t}}{\alpha_t}} X_t + (\sqrt {1-\alpha_{t-\Delta t}}+ \sqrt{\frac{\alpha_{t-\Delta t}(1-\alpha_t)}{\alpha_t}}) \epsilon_\theta^t(X_t)$. Taking the Jacobian of $D_g(X_t)$ with respect to $X_t$ yields: $D_{g}(X_t) = \sqrt{\frac {\alpha_{t-\Delta t}}{\alpha_t}} I + (\sqrt {1-\alpha_{t-\Delta t}}+ \sqrt{\frac{\alpha_{t-\Delta t}(1-\alpha_t)}{\alpha_t}}) D_{\epsilon_\theta^t}(X_t)$, where $I$ is the identity matrix. Thus, applying $D_g(X_t)$ to a vector $v$ produces Eq.8.
>
> **Q4. Clarification on Perceptual Metrics and Generation Rate.**
>
> The generation rate at data point level $X_t$ makes sense since it can also be interpreted as the rate at which noise is removed as $X_t$ moves closer to the original image $X_0$. In practice, this is effectively captured by the perceptual distance change computed through metrics such as LPIPS between the estimated $\hat X_0(X_t)$ and the real image $X_0$.
>
> **Q5. Obtaining Reference Generation Curves via Image Inversion.**
>
> Yes, we use diffusion ODE inversion to obtain generation curves for real images. We will clarify this clearly in Section 3.
>
> **Q6. Leveraging Generation Curves for Image Editing Applications.**
>
> The positive correlation of saliency with curve fluctuation is leveraged as follows:
>
> - **Saliency Manipulation and Blending:** By enhancing or suppressing the curve fluctuation of pixels, we directly control their saliency; disharmonious object boundaries, identified as inherently salient regions, are smoothed by reducing their curve fluctuations.
>
> - **Semantic Transfer and Object Removal:** The overall shape of curves encodes broader visual semantics, as discussed in lines 204-210. We transfer these semantics by matching the entire shape of generation curves; object removal is treated as a special case where semantics are shifted from the object to its background.
>
> These methods can be applied to arbitrary image regions, which do not necessarily have to be salient.

---

> > ### Comment · Reviewer_SNns · 2025-04-06
> >
> > I appreciate the authors for addressing my questions. Since the concerns raised in my original review have been resolved, I will increase my rating.

---

> > > ### Author Response · Authors · 2025-04-08
> > >
> > > Thank you for your positive feedback. We are glad that our revisions addressed your concerns, and we appreciate your updated rating.

---

### Official Review · Reviewer_xSY4 · 2025-03-12

**Overall Recommendation:** 3

**Summary:**

This work researches the visual properties of images during the diffusion process. By employing the manifold hypothesis, the authors propose a new metric called the generation rate. They experimentally show correlation of this metric with the visual properties of image generation. Furthermore, the authors design an algorithm that can manipulate the visual properties of image generation by matching the generation rate during the diffusion process.

**Claims And Evidence:**

- In Line 154-157, are there any theoretical results that show $D_{f_t}$ is contractive?
- In Line 242, what do you mean "... dominates the overall shape ..."? And for figure 5, I cannot find the yellow curve.

**Essential References Not Discussed:**

No.

**Experimental Designs Or Analyses:**

Yes,

**Methods And Evaluation Criteria:**

Yes.

**Other Comments Or Suggestions:**

- I thought it was better to provide more preliminaries and notations of manifold in Appendix A. For example, the ODE $dx = f(x,t)dt$ with initial value $x(0) = x_0$ induces the flow $\{\Phi_t\}$ with $\Phi_0(x_0)=x_0$ and $\Phi_t(x_0) = x(t)$. So in this paper, $M_t = \Phi_t(M_0)$, where $M_0$ is the target data manifold, and $f_t = \Phi_{\Delta t} \colon M_{t-\Delta t} \colon \rightarrow M_t$ is a diffeomorphism. I thought it may be a better way to understand the concept of time-varying manifolds.

- Based on the notations in Appendix A, in the main part of this paper, I think $D_{f_t}$ and $D_{f_t^{-1}}$ should be replaced by $D_xf_t$ and $D_xf_t^{-1}$ respectively and so $D_{f_t^{-1}}(X_t)[v]$ should be $D_{X_t}f_t^{-1}(v)$.

- For the Section 2.2, because the power method and $h_t$ has been widely used in the following contents, I thought it is better to provide more details of these techniques in appendix part.

**Other Strengths And Weaknesses:**

### Strength:
- They define a new metric, generation rate, for measure the rate of the change of the data manifold in the diffusion process. They empirically show that this new rate has strong connections with the rate of the information removal during the diffusion process.
- By manipulating the generation rate, they propose a matching algorithm that can modify the visual properties of generated image, which is an interesting technique. They show this technique can be applied to different tasks of image manipulation.



### Weakness:
- The motivation of defining the generation rate as the norm of differential map seems to be insufficient, except for experimental results.
- There are some explanations that are too experimental, like replacing $r_t$ by $\left\Vert D_{\epsilon_\theta^t}\left(X_t\right)[\operatorname{Proj}(v)]\right\Vert $ and replacing $\left\Vert D_{\epsilon_\theta^t}\left(X_t\right)[\operatorname{Proj}(v)]\right\Vert $ by $\left\Vert D_{h_t}\left(X_t\right)[v]\right\Vert $. It is better to provide more rigorous analyses.

**Questions For Authors:**

- In Line 121, about the projection operator $\text{Proj}(v)$, can we explicitly provide the formula for calculating $\text{Proj}(v)$ when given a $v \in \mathbb{R}^d$ and whether is this method computationally efficient?

**Relation To Broader Scientific Literature:**

This work proposes to research some metrics during the diffusion process, which may have strong connections with the visual properties of images. Although it lacks rigorous mathematical explanations, the experimental results show the potential of this view that can have wide applications.

**Theoretical Claims:**

Yes.

---

> ### Author Rebuttal · Authors · 2025-04-01
>
> Thank you for your detailed review! Below we address the questions.
>
> **Q1.Theoretical Justification of Contractive $D_{f_t}$.**
>
> The observation of contractility is primarily empirical, demonstrated through experiments where tangent vectors {$v_i$} are scaled under the forward differential $D_{f_t}[v]$. This empirical contractility serves to intuitively illustrate our idea rather than being a necessary condition for defining the generation rate. Even without it, the scaling factor can still meaningfully quantify generation rates.
>
> **Q2. Clarification of 'Dominates Overall Shape' and Visibility of Yellow Curve in Fig.5.**
>
> We clarify the words 'dominates the overall shape' as indicating that the shape of $\|D_{\epsilon_\theta^t}(X_t)[\text{Proj}(v)]\|$ closely resembles the shape of the original $\|D_{f_t}(X_t)[\text{Proj}(v)]\|$. Regarding the visibility of the yellow curve in Figure 5, it overlaps significantly with the red curve in the first plot due to their high similarity, particularly visible upon closer inspection in the timestep range $t\in [0,40]$.
>
> **Q3. Expanded Preliminaries and Manifold Notations.**
>
> Following your suggestion, we will revise Appendix A to explicitly introduce the notations of diffeomorphism induced by the diffusion ODE. We agree to replace $D_{f_t}$ consistently with the clearer notation $D_xf_t$ throughout the paper.
>
> **Q4. Additional Details of Power Method in Appendix.**
>
> The power method and the identification of $h_t$ as a contractive map are discussed extensively in [Park2023], specifically in their Appendix F. We will expand our Appendix to briefly summarize these foundational details.
>
> **Q5. Formula and Computational Efficiency of $\text{Proj}(v)$.**
>
> Given an orthonormal tangent basis {$u_i$}$^k_{i=1}$ and an arbitrary ambient-space vector $v \in R^n$, the projection operator can be explicitly expressed as: $\text{Proj}(v) = UU^Tv$, where $U=[u_1...u_k]$ is an $n \times k$ matrix of tangent vectors. The above matrix multiplication is efficient, with primary complexity arising from calculating the tangent basis. Using the power method, this procedure typically requires tens of seconds on a GPU. We note this complexity is inherently necessary due to the high dimensionality of the ambient space.
>
> [Park2023]Park et al.. Understanding the latent space of diffusion models through the lens of riemannian geometry. NeurIPS, 2023.

---

### Official Review · Reviewer_xDCq · 2025-03-17

**Overall Recommendation:** 3

**Summary:**

Motivated by differential geometry of manifold, the authors defined a metric called (projected) generation rate. It signifies how much a direction on tangent space of the manifold at time t were amplified or diminished through the reverse diffusion mapping. They manage to compute it locally for a patch of pixels. Then, they empirically found the generation curve fluctuation perceptually correlates with the visual saliency of the patch. They developed a method to optimize / control the generation curve, with several applications e.g. semantic transfer, object removal, saliency manipulation and image blending.

All in all they discovered an interesting geometric aspect in the diffusion generation process, and found interesting image control usage of it.


## Update after rebuttal
The authors clarified the theoretical links to a few previous papers around diffusion sampling and showed the consistency with their observation. They also showed stronger statistical tests of their observation.
The reviewer is happy to retain the score.

**Claims And Evidence:**

Mostly yes! See below.

**Essential References Not Discussed:**

See above.

**Experimental Designs Or Analyses:**

- **Relation of saliency and generation curve fluctuation**
In figure 2 the caption mentioned “*The generation curves fluctuate significantly at the pixels with high visual saliency, such as the wing tip of the bird.*” Though the visual looks really cool, it’s not visually clear how the example on left side reflects this claim. Is there any quantification showing the correlation between them?
    - For visual saliency there are some computational surrogates i.e. DNN models trained to predict saliency in perception.
- **Quantification of the relation between saliency and curve fluctuation**
“*For 86% of the images, higher visual saliency leads to higher fluctuation, validating the high consistency between curve fluctuation and visual saliency.*” For this results in Figure 3 left, can we plot the results in a more salient way? since the image sameples do not have an order, you can even plot the salient point variance against the non salient point variance.
Further saliency is a continuous thing, and do not need to be thresholded here. Could you report the correlation between saliency value and windowed variance of the curve? Generally, I like the results in 3.2 for connection of generation curves to visual saliency, but I think better statistical quantification of their correlation could be useful and more convincing.

**Methods And Evaluation Criteria:**

- **Minor point about the design of current method:**
It seems from the current set up (Eq. 6), the projected generation rate is computed from the discretized sampling equation 4. As we know from EDM line of work, the time parametrization is somewhat arbitrary and could change for different scheduler. Even changing the sampling step size will change it. So would it be more proper to build the method on continuous time formulation (Eq. 3) which quantifies the instantaneous generation rate. Then it will be time scheduling independent and easier to compare across models?
    - I feel a more general / continuous time version of this work would be to compute the Jacobian of the end state as a function of the initial state, through the integration of the ODE.
- **Major point about the method : Manifold projection**
“*To mitigate this flaw, we define the projection operator Proj(v) as the projection of v onto the tangent space spanning by the leading singular vectors,*” I love the idea of finding on manifold directions. Though I’m not sure how critical the choice is.
    - Theoretically speaking, early in diffusion, as you have plotted in Figure 1, the noised data distribution is very close to Gaussian, so the directions should be isotropic and I’m not sure there is any special “on-manifold” direction. So the search for projection might not add value to the metric.
    - Empirically speaking, could there be an ablation study about the choice for the projection operation? How important it is to use the encoder layer of UNet to help? It seems to make the method very dependent on UNet based architecture. I guess, some other choices e.g. dataset PCA / PCA of trajectory can provide a principled way for projecting the perturbations. C.f. [WV2023] Fig. 18. Current method of using “power law based derivation of the tangent space” seems expensive and not sure how crucial it is.
- From the description in Sec4.2, the optimization procedure of Eq. 10 is not totally clear to me.  “*We update Xt using the gradient descent step Xt ← Xt−η∇Xt |c(ts|X , eij)−c⋆(ts)| with η being the learning rate of the SGD optimizer. After optimization, we recover .X0 from the optimized .Xt by Equation 4.*”
Since the sampling trajectory is obtained via discretizing ODE, the states have sequential dependency. Do you optimize one time point at a time? with randomly chosen time points? do you need to traverse the sampling procedure multiple times to perform multiple gradient steps? Authors could consider make it clearer during revision (e.g. making an algorithm box)

[WV2023] Wang, B., & Vastola, J. J. (2023). Diffusion models generate images like painters: an analytical theory of outline first, details later. *arXiv preprint arXiv:2303.02490*.

**Other Comments Or Suggestions:**

Suggestions for style

- For many line plots showing generation rate / curve, the legend and xy axes annotations are too small, could be made larger for easier reading.
- Missing link around L873
- Mis statement: “*using only a pre-trained, unconditional diffusion model for image generation.*” Authors used SD2.1-base which is a conditional diffusion model, i.e. image sampling conditional on text.

**Other Strengths And Weaknesses:**

**Strength**

- I think the approach and theoretical framework presented in the paper is very novel and creative.
- The comparison with the other generation rate / curve estimation method (Choi et al 2022) is interesting, showing they may be estimating similar underlying things. The previous method seems more intuitive and designed to be perceptually aligned.

**Questions For Authors:**

- Is there some intuitive explanations of why the generation rate should connect with curve fluctuation? I’m very curious!

**Relation To Broader Scientific Literature:**

- I think the direction of leveraging generative models to study data geometry is super interesting, and can possibly reveal aspects of natural image manifolds that are hidden before. At the time of GAN and VAEs, quite a few groups tried to understand the image space / latent space geometry using similar motivations, i.e. understanding the Riemannian metric tensor of the latent space. [AHH2017] [SKT2018] [WP2021] [CA2022]
- One paper that shares notable conceptual similarity to the current one is [WP2021], which also traced the change of manifold, throughout the layers in the GAN network, using differential geometry language. Basically, they defined a similar quantity like rate of image change when traveling along different directions locally. Luckily for GAN the tangent space is defined by the latent space, so no need to search for it.

[AHH2017] Arvanitidis, G., Hansen, L. K., & Hauberg, S. (2017). Latent space oddity: on the curvature of deep generative models. ICLR

[SKT2018] Shao, H., Kumar, A., & Thomas Fletcher, P. (2018). The riemannian geometry of deep generative models. In *Proceedings of the IEEE Conference on Computer Vision and Pattern Recognition Workshops* (pp. 315-323).

[WP2021] Wang, B., & Ponce, C. R. (2021). The geometry of deep generative image models and its applications. ICLR

[CA2022] Chadebec, C., & Allassonnière, S. (2022). A geometric perspective on variational autoencoders. NeurIPS

**Theoretical Claims:**

N.A.

- I feel the paper might benefit from some theoretical and conceptual framing of what the generation curve / rate represents and what the expansion rate means. ( I know it correlates with visual saliency, but I kind of want a more principled explanation of their relation )
    - I feel the authors might refer to [WV2023] In their case (e.g. approximating image manifold with Gaussian), then the on manifold directions are well defined and the generation rate can be computed analytically. Basically, on manifold perturbations would be amplified at varying rate, and off manifold perturbations will shrink through reverse diffusion.  [WV2023] Fig.22 showed that empirically.
    - In [WV2023] the authors also observed the different kind of elements specified at different point of the generation process, e.g. layout / low frequency / high variance elements specified first, object details specified later. So I guess there is some connection to the shape the the temporal position of the peaks in the current work. (Fig2 right)

[WV2023] Wang, B., & Vastola, J. J. (2023). Diffusion models generate images like painters: an analytical theory of outline first, details later. *arXiv preprint arXiv:2303.02490*.

[WV2024] Wang, B., & Vastola, J. J. (2024). The Unreasonable Effectiveness of Gaussian Score Approximation for Diffusion Models and its Applications. TMLR

---

> ### Author Rebuttal · Authors · 2025-04-01
>
> Thank you for your detailed review! Below we address the questions.
>
> **Q1. Continuous-Time Formulation of Generation Rate**
>
> We agree that leveraging a continuous-time formulation provides a scheduler-independent definition. For the diffusion ODE $\frac d {dt}X_t=h(X_t,t)$, the variation of the norm of a unit tangent vector $v_t$ can be expressed as $\frac d {dt}\|v_t\|= \left \langle v_t, D_{h(X_t,t)}[v_t]\right \rangle$. We will clearly present and discuss this formulation in Section 3.
>
> **Q2-1. Necessity of Projection onto Tangent Space**
>
> In diffusion ODE, the data manifold $M_0$ is continuously transformed into $M_t=\phi_t(M_0)$ by induced diffeomorphism $\phi_t$, preserving a low-dimensional manifold structure at all times. Therefore, the projection onto $T_XM_t$ remains necessary. Although ideally one can track the $T_XM_t$ by pushing forward $T_XM_0$ via the diffusion ODE, under computational constraints we approximating the tangent space at each timestep. This is reliable at moderate to low noise levels despite some loss of precision at very high noise levels.
>
> **Q2-2. Alternative Projection Methods**
>
> Dataset PCA is impractical because it requires full dataset access at each noise level {$X_t^i$}$_{i=1}^N$, and PCA on single trajectories (as in [WV2023]) yields overly restricted subspaces. The adopted power-based approach, while dependent on the UNet architecture, is a practical solution. If there are other viable methods for high-dimensional nature image analysis, we will incorporate ablation studies with them.
>
> **Q3. Clarification on Optimization Algorithm**
>
> At each iteration, for variable $X_t$, we samples a random timestep $t_s$ and traverse the diffusion ODE from $X_t$ to $X_{t_s}$ without gradients. We then calculate gradients at $X_{t_s}$ and propagate them back to $X_t$ using adjoint method incorporated in PyTorch. We will clarify these details further in Section 4.2 and in the algorithm box.
>
> **Q4. Theoretical Explanation Connecting Generation Curve and Visual Saliency**
>
> - **Critical Period and Peak Positions** According to [WV2024], the estimated $\hat{X_0}$ undergoes a critical period of rapid change. Since the rate of change of $\hat{X_0}$ parallels our generation rate (as discussed in Section 3.2), this period directly corresponds to the dominant peak in our generation curves. The relation between critical period and feature variance[WV2024] also explains how peak positions encode detailed visual semantics essential for our semantic transfer task.
>
> - **Non-salient Curves with Peaks Near $t=0$**: [WV2024] notes that high-frequency features tend to have lower variance, causing their critical periods to occur later in the diffusion process. For non-salient backgrounds (like a plain wall), low-frequency components are largely captured by the distribution mean approximated by $\hat{X_0}(X_T)$. Consequently, only the high-frequency details—such as subtle textures—require explicit generation, which consistently results in peaks near $t=0$.
> Moreover, [WV2024] Figure 3.D analytically demonstrates that when the critical period occurs near zero, the corresponding change rate surges sharply, a result that aligns with our empirical findings (lines 191–194) of non-salient curves.
>
> -  **Curvature and saliency**: Considering the continuous diffusion ODE expressed by score function $dX_t = -g(t)s(X_t,t)dt$, the generation rate can be related to the Hessian of the log probability density: $r(X_t,v) = |g(t) \left \langle D_{s(X_t,t)}[v], v \right \rangle | = |g(t) \left \langle D_{\nabla logP(X_t)}[v], v \right \rangle |= |g(t) \text{Hessian}_{log P}(v,v) |$. Intuitively, the Hessian measures curvature-the sensitivity to perturbations in a given direction. Visually salient areas, rich in semantic details, may exhibit larger curvature: For example, the change of boundary pixels can disrupt the outline of an object, resulting in the deviation from image manifold. This geometric perspective explains why salient regions have greater fluctuations.
>
> **Q5. Saliency and Curve Fluctuation Experiments**
>
> - **Visual examples (Figure 2 left):** We provide saliency maps (via EML-NET) in https://limewire.com/d/3h4XH#SSSDAgZ0rL Figure 1; Figure2 left highlights salient areas such as the mantis and the center of the lotus leaf, with additional curve fluctuations from leaf veins.
>
> - **Statistical correlation:** We first converted the discrete salient points in dataset to continues maps via Gaussian blurring as suggested. Statistical tests conducted on dataset yields:
>
> - Point-Biserial: $(r = 0.297, p=1.36e^{-69})$
>
> - t-test: $(t = 18.050, p=1.36e^{-69})$
>
> These confirm a strong positive correlation between fluctuation and saliency. This evaluation was conducted on an expanded dataset of 2000 images.
>
> **Q6. Suggestions for Style.**
>
> We will adjust the plots and update the missing link as suggested. We refer to our model as 'unconditional' because we exclusively use its unconditional mode (with text prompt 'None').

---

> > ### Comment · Reviewer_xDCq · 2025-04-07
> >
> > We appreciate the authors’ efforts to connect their current method with other theoretical frameworks and previous results on the diffusion sampling process (e.g., [WV2024]). This integration significantly enhances the overall presentation of their new perspective.
> > Furthermore, we commend the authors for providing additional clarifications on the method and for the more robust statistical quantification of the results. Accordingly, we will maintain our score as is!

---

> > > ### Author Response · Authors · 2025-04-08
> > >
> > > We thank the reviewer for the positive and thoughtful feedback. We are pleased that the integration with [WV2024] and additional clarifications have enhanced the presentation of our work. Your supportive comments are greatly appreciated.

---

### Decision · Program_Chairs · 2025-05-01

**Decision:**

Reject

**Comment:**

The draft proposes a new function for tracing the change of the data manifold as it evolves through diffusion layers and there are two main claims about it; (i) that the function correlates with visual features e.g. saliency, and (ii) that the function has wide applicability in a large range of downstream tasks.
Based on the reviews and my own reading on the paper, I was left with a bittersweet taste of a paper that could have been accepted (there is unanimous agreement that the work is interesting and has large potential -acknowledging support for claim 2-, but just fall short of being convincing by lacking both any theoretical support and even the empirical evidence not being supported by statistics and being limited to qualitative appreciations.
All reviewers acknowledged the rebuttal and 2 of them even increased their initial score from weak reject (2) to weak accept (3) but none were a clear acceptance. The rebuttal itself covers all main points mentioned by the authors but does so in a similar tone to the main paper mostly staying in the qualitative side.
Overall, I can see how this paper could be flying in other forums, but it just feels that ICML is not the right one for it with the current approach to communicating the work.